

# Natural ocean acidification at Papagayo upwelling system (North Pacific Costa Rica): implications for reef development

Celeste Sánchez-Noguera[1,2], Ines Stuhldreier[1,3], Jorge Cortés[2], Carlos Jiménez[4,5], Álvaro Morales[2,6], Christian Wild[3], Tim Rixen[1,7]

[1]Leibniz Centre for Tropical Marine Research (ZMT), Bremen, D-28359, Germany
[2]Centro de Investigación en Ciencias del Mar y Limnología (CIMAR), San José, 11501-2060, Costa Rica
[3]Faculty of Biology and Chemistry (FB2), University of Bremen, Bremen, 28359, Germany
[4]Energy, Environment and Water Research Center (EEWRC) of the Cyprus Institute (CyI), Nicosia, 1645, Cyprus
[5]Enalia Physis Environmental Research Centre (ENALIA), Aglanjia, 2101, Nicosia, Cyprus
[6]Escuela de Biología, University of Costa Rica, San José, Costa Rica
[7]Institute of Geology, University Hamburg, Hamburg, 20146, Germany

*Correspondence to*: Celeste Sánchez-Noguera (celeste08@gmail.com)

**Abstract.** Numerous experiments have shown that ocean acidification impedes coral calcification, but knowledge about in situ reef ecosystem response to ocean acidification is still scarce. Bahía Culebra, situated at the northern Pacific coast of Costa Rica, is a location naturally exposed to acidic conditions due to the Papagayo seasonal upwelling. We measured pH and $p$CO$_2$ in situ during two non-upwelling seasons (June 2012, May-June 2013), with a high temporal resolution of every 15 and 30 min, respectively, using two Submersible Autonomous Moored Instruments (SAMI-pH, SAMI-CO2). These results were compared with published data from the upwelling season 2009. Findings revealed that the carbonate system in Bahía Culebra shows a high temporal variability. Incoming offshore waters drive inter- and intra-seasonal changes. Lowest pH (7.8) and highest $p$CO$_2$ (658.3 µatm) values measured during a cold-water intrusion event in the non-upwelling season were similar to those minimum values reported from upwelling season (pH = 7.8, $p$CO$_2$ = 643.5 µatm), unveiling that natural acidification occurs sporadically also in non-upwelling season. This affects the interaction of photosynthesis, respiration, calcification, and carbonate dissolution and the resulting diel cycle of pH and $p$CO$_2$ in the reefs of Bahía Culebra. During non-upwelling season, the aragonite saturation state ($\Omega_a$) rises to values of >3.3 and enhances calcification. Aragonite saturation state values during upwelling season falls below 2.5, hampering calcification and coral growth. Low reef accretion in Bahía Culebra indicates high erosion rates and that these reefs grow on the verge of their ecological tolerance. The $\Omega_a$ threshold values for coral growth, derived from the correlation between $\Omega_a$ and coral linear extension rates, suggest that future ocean acidification will threaten reefs in Bahía Culebra. These data contribute to build a better understanding of the carbonate system dynamics and coral reefs key response (e.g. coral growth) to natural low-pH conditions, in upwelling areas in the Eastern Tropical Pacific and beyond.





## 1 Introduction

Ocean acidification (OA) caused by human-induced increase of atmospheric $CO_2$ (Sabine et al., 2004; Feely et al., 2009) is considered one of the major threats to marine calcifying organisms and ecosystems (Fabry et al., 2008; Hofmann et al., 2010; Doney et al., 2012; Gattuso et al., 2015). Among all marine habitats, tropical coral reefs are recognized as the most

endangered ones (Hoegh-Guldberg et al., 2007; Kleypas and Yates, 2009; Pörtner et al., 2014), since in addition to reduced calcification (Langdon et al., 2000; Marubini et al., 2008; Doney et al., 2009; Gattuso et al., 2014) a lower pH also weakens the reef framework by favoring bioerosion and enabling carbonate dissolution (Gattuso et al., 2014; Manzello et al., 2014; Barkley et al., 2015). According to the IPCC business-as-usual scenario, about 90% of the ocean's surface waters will become undersaturated with respect to aragonite in the next decades (Gattuso et al., 2015), emphasizing the need to study the

response of natural ecosystems to OA. Nowadays, aragonite undersaturated surface waters occur naturally in some parts of the ocean, as consequence of underwater volcanic seeps (Hall-Spencer et al., 2008; Fabricius et al., 2011; Enochs et al., 2015; Fabricius et al., 2015) or upwelling that drags corrosive deep water into the surface mixed layer (Feely et al., 2008; Hauri et al., 2009; Fassbender et al., 2011; Harris et al., 2013).

AAside from some studies at volcanic seeps (Fabricius et al., 2011; Kroeker et al., 2011; Enochs et al., 2015; Fabricius et al.,

2015) or at reefs in the Eastern Tropical Pacific (ETP) (Manzello, 2008, 2010a, 2010b; Manzello et al., 2008, 2014), our understanding of OA impacts on corals derives mainly from laboratory and seawater enclosure experiments (Pörtner et al., 2014; Hall-Spencer et al., 2015). These results are used to predict ecosystem responses to future OA (Kleypas et al., 2006; Kleypas and Langdon, 2006), but their reliability is challenged by the artificial conditions under which the experiments are conducted. For example, the duration of studies is often too short to allow a full adaptation of the organisms/systems to the

changing environmental conditions, and the missing connectivity between ecosystems in seawater enclosures restricts natural interactions between organisms (Kleypas et al., 2006; Kleypas and Langdon, 2006; Hofmann et al., 2010). In situ studies in natural low-pH conditions are able to overcome some of these problems and the ETP is well known for its $CO_2$-enriched and acidic subsurface waters (Takahashi et al., 2014). Upwelling events decreases the carbonate saturation state ($\Omega$) along the Central American coast (Manzello et al., 2008; Manzello, 2010b; Rixen et al., 2012), resulting in poorly cemented coral reefs

with low accretion rates that are subject to rapid bioerosion (Manzello et al., 2008; Alvarado et al., 2012).

Corals in the northern part of the Costa Rican Pacific coast are developing under the influence of the seasonal Papagayo upwelling (Jiménez et al., 2010; Rixen et al., 2012; Stuhldreier et al., 2015a, 2015b). To contribute to the general understanding of OA impacts on coral reefs, we investigated the variability of the carbonate system in the upwelling-influenced Bahía Culebra, Costa Rica. The main objectives of this study were 1) to describe the behavior of the carbonate

system on diurnal and seasonal time scales, 2) to characterize the controlling processes, and 3) to determine ecological impacts of changing carbonate systems. Furthermore, our results will allow us to draw some conclusions concerning future thresholds of coral survival.



## 2 Methods

### 2.1 Study site

Bahía Culebra, located in the Gulf of Papagayo, North Pacific coast of Costa Rica (Fig. 1), is strongly influenced by the northeasterly Papagayo winds. The strongest wind jets develop during the boreal winter (Amador et al., 2016) and are driven

by large-scale variations of the trade winds (Chelton et al., 2000; Alfaro and Cortés, 2012). When Papagayo winds blow through the mountain gap between southern Nicaragua and northern Costa Rica, the resulting strong offshore winds on the Pacific side lead to upwelling of cold and nutrient-enriched subsurface waters between December and April (McCreary et al., 1989; Brenes et al., 1990; Ballestero and Coen, 2004; Kessler, 2006;). These cyclonic eddies also influence the magnitude and location of the Costa Rica Dome (CRD), which is located ca. 300 km off the Gulf of Papagayo (Fiedler, 2002).

However, the CRD changes its distance to the Costa Rican coast throughout the year, as a result of differences in wind forcing (Wyrtki, 1964; Fiedler, 2002). During the dry season, particularly between February and April, offshore moving water masses strengthen upwelling at the coast and shoal the thermocline in the Gulf of Papagayo (Wyrtki, 1965, 1966; Fiedler, 2002). In May-June, during the onset of the rainy season the CRD moves offshore (Fiedler, 2002; Fiedler and Talley, 2006) and the North Equatorial Countercurrent (NECC) can carry tropical water masses into Bahía Culebra until

December, when again upwelling sets in (Wyrtki, 1965, 1966).

### 2.2 Measurements

We measured in situ pH, $pCO_2$ and seawater temperature (SWT) during two non-upwelling periods (15 days in June 2012 and 7 days in May-June 2013, Fig. 2). Measurements were undertaken with two Submersible Autonomous Moored Instruments (SAMI-pH and SAMI-CO$_2$) (www.sunburstsensors.com), in sampling intervals of 15 (June 2012) and 30

minutes (May-June 2013). SAMI-sensors were deployed at the pier of Marina Papagayo (85°39'21.41"W; 10°32'32.89"N), on top of a carbonate sandy bottom in the inner part of Bahía Culebra (Fig. 1). The water-depth varied approximately between 5-8 m depending on the tide, but sensors, hooked to the pier, moved up and down with the tide and were always at the same depth, 1.5 m below the surface. SAMI instruments measured pH and pCO$_2$ spectrophotometrically by using a colorimetry reagent method (DeGrandpre et al., 1995, 1999; Seidel et al., 2008). Salinity from discrete samples was

measured with a WTW probe (Cond3310) and used for correction of pH values. Calculation of aragonite saturation state ($\Omega_a$) from parameters measured in situ with SAMI sensors is accurate (Cullison Gray et al., 2011; Gray et al., 2012), but discrete water samples were collected as often as possible to validate the instruments (Fig. 3). 250 mL borosilicate bottles were filled with seawater at 30 cm below the surface and preserved with 200 μl of 50% saturated HgCl$_2$ solution to inhibit biological activity (Dickson et al., 2007). Samples were stored at 3-4 °C until analysis. Total alkalinity (TA) and Dissolved inorganic

carbon (DIC) were measured using a VINDTA 3C (Versatile Instrument for the Determination of Total dissolved inorganic carbon and Alkalinity; Marianda, Kiel, Germany) coupled with a UIC CO$_2$ coulometer detector (UIC Inc., Joliet, USA). Both instruments were calibrated with Dickson Certified Reference Material (Batch 127) (Dickson et al., 2003). DIC




concentrations as well as TA and $\Omega_a$ were calculated with the CO2SYS program as a function of measured pH and $p$CO$_2$; with dissociation constants of Mehrbach et al. (1973) for carbonic acid as refit by Dickson and Millero (1987), and Dickson (1990) for boric acid.

Wind speeds were obtained from a station of the Instituto Metereológico Nacional (National Metereological Institute of Costa Rica), located at the nearby Liberia airport. The Módulo de Información Oceanográfica of the University of Costa Rica (www.miocimar.ucr.ac.cr) supplied the tidal data.

## 2.3 Data analysis

We compared our data with values measured during upwelling season in 2009 (Rixen et al., 2012). Correlations between tidal cycles and physicochemical parameters (pH, $p$CO$_2$, T, wind) during non-upwelling periods were tested via Pearson Correlation in Python. Differences in parameters (temperature, pH, $p$CO$_2$, TA, DIC, $\Omega_a$) between all periods (2009, 2012, 2013) were tested with a General Linear Model (GLM), in the statistical package R. The GLM was evaluated using graphical methods to identify violations of assumptions of homogeneity of variance and normality of residuals. Additionally, we developed a simple model to improve our understanding of processes controlling the observed diel trends, as seen in the time series data of pH and $p$CO$_2$ (Fig. 2, 4). The model simulates combined effects of metabolic processes (photosynthesis, respiration, calcification and dissolution) on the carbonate chemistry. Input parameters for starting the model were DIC and TA values corresponding to the highest and lowest measured $p$CO$_2$ during day and night. The difference between the two DIC concentrations ($\Delta$DIC) was assumed to be caused by photosynthesis/respiration and the resulting formation and decomposition of particulate organic carbon (POC) as well as calcification/dissolution and the precipitation and dissolution of particulate inorganic carbon (PIC, Eq. 1). The rain-ratio ($R_{OI}$=POC/PIC) was used to link $\Delta$POC to $\Delta$PIC (Eq. 2, 3). The rain ratio was further constrained by the determined change of TA ($\Delta TA$). Therefore, it was considered that photosynthesis/respiration of one mole of carbon increases and reduces TA by 0.15 units respectively (Broecker and Peng, 1982). Calcification/dissolution of one mole of carbon decreases and increases TA by two units (Eq. 4). To obtain the best fit between the determined (measured) and the calculated ($\Delta TA$) a $R_{OI}$ of -2.6 and 0.8 was used for the year 2012 and 2013, respectively. To further verify the model results the calculated $\Delta$DIC and $\Delta$TA were used to calculate $p$CO$_2$ and pH, which were compared to the measured ones (Fig. 5).

$$\Delta DIC = \Delta POC + \Delta PIC \qquad (1)$$

$$\Delta PIC = \left(\frac{\Delta POC}{R_{OI}}\right) \qquad (2)$$

$$\Delta POC = \Delta DIC / \left(1 + \left(\frac{1}{R_{OI}}\right)\right) \qquad (3)$$

$$\Delta TA = (\Delta POC * 0.15) - \left(\left(\Delta \frac{POC}{R_{OI}}\right) * 2\right) \qquad (4)$$




This was calculated on hourly time steps, separately for 2012 and 2013, using the mean SWT (2012 = 29.61 ± 0.93 °C, 2013 = 30.08 ± 0.27 °C) and salinity (2012 = 32.5 psu, 2013 = 32.5 psu).

## 3 Results

### 3.1 Carbonate chemistry during non-upwelling season

In June 2012, average SWT was 29.61 ± 0.93 °C and ranged from 27.13 °C to 31.37 °C. In May-June 2013 SWT ranged from 29.3 °C to 30.7 °C (average 30.08 ± 0.27°C). During both periods, the salinity was 32.5 ± 0.8 psu. During the study periods, the wind intensified during the afternoons reaching speeds of up to 8.5 m s$^{-1}$ and 6.0 m s$^{-1}$ in 2012 and 2013, respectively (Fig. 2). Average pH and $p$CO$_2$ in June 2012 were 7.98 ± 0.04 and 456.38 ± 69.68 µatm, respectively; the corresponding averages for May-June 2013 were 8.02 ± 0.03 and 375.67 ± 24.25 µatm. Since the tidal cycle did not significantly correlated with the variability of pH, $p$CO$_2$, T or wind ($p > 0.05$) during the periods of observations (Table 2), it was excluded from further discussions. Mean $\Omega_a$ values were 3.32 ± 0.46 in June 2012 and 3.50 ± 0.49 in May-June 2013 (Table 1).

### 3.2 Seasonal variation of the carbonate system

All GLM assumptions were met and measured parameters showed significant differences between study periods ($p < 0.05$). The SWT range differed between years (Table 1); 2013 was the warmest study period, followed by 2012 and 2009. Lowest measured pH was 7.81 in June 2012, 7.84 in April 2009 and 7.95 in May-June 2013. To minimize the temperature effect, we compared DIC and TA instead of $p$CO$_2$. Highest DIC values were 2360.92 µmol kg$^{-1}$ in April 2009, 2355.39 µmol kg$^{-1}$ in June 2012 and 2199.50 µmol kg$^{-1}$ in May-June 2013. Similarly, highest TA values were 2714.84 µmol kg$^{-1}$ in June 2102, 2610.47 µmol kg$^{-1}$ in April 2009 and 2599.99 µmol kg$^{-1}$ in May-June 2013. According to average values, April 2009 was the period with most acidic water and greater CO$_2$ enrichment, followed by June 2012 and May-June 2013 (Table 1). Mean $\Omega_a$ values were 2.71 ± 0.29 during upwelling season (April 2009) and 3.37 ± 0.47 during non-upwelling season (June 2012, May-June 2013), resulting in an annual average $\Omega_a$ of 3.04 at Bahía Culebra. Time series of pH and $p$CO$_2$ in June 2012 and May-June 2013 showed a pronounced daily cycle (Fig. 4), which in addition to previously described data will be discussed in the following paragraphs.

## 4 Discussion

### 4.1 Natural OA beyond the upwelling season

The observed differences in pH and $p$CO$_2$ between 2012 and 2013 suggest that the non-upwelling season exhibits a strong interannual variability (Table 1). In 2012 pH was lower and $p$CO$_2$ higher than in 2013 (Fig. 2b, c). The June 2012 time-series data showed that SWT decreased and $p$CO$_2$ increased from 300 to 650 µatm in less than a week, after several days of strong afternoon winds (Fig. 2a). This suggests, that an enhanced wind-driven vertical mixing entrained cooler and CO$_2$-enriched





waters from greater water-depth into the surface layer. The associated SWT drop from 31.4 °C to 27.1 °C was similar to those observed during the onset of the 2009 upwelling event (26.2 °C to 23.7 °C; Rixen et al., 2012). Nevertheless, the higher SWT during the 2012 non-upwelling season suggests that the entrained water originated from a shallower water-depth, compared to the water upwelled in 2009. The $p\mathrm{CO_2}$ values with up to 650 µatm reached the same level during both

events, which is partially caused by the higher SWT in 2012. However, the temperature independent DIC concentrations in 2012 (1924.65 ± 195.07 µmol kg$^{-1}$) fell below those in 2009 (2098.71 ± 103.81 µmol kg$^{-1}$), but exceeded those in 2013 (1800.92 ± 142.78 µmol kg$^{-1}$, Table 1). This implies that in addition to high SWT, the entrainment of $\mathrm{CO_2}$-enriched subsurface water increased the $p\mathrm{CO_2}$ not only during the upwelling periods, but also during the 2012 non-upwelling season. Since in 2012 the $p\mathrm{CO_2}$ increased already in June 7$^{th}$ and the SWT decreased only two days later (June 10$^{th}$), the inflow of

$\mathrm{CO_2}$-enriched waters seems to have increased the $p\mathrm{CO_2}$ already prior to the strengthening of local winds (Fig. 2b). Later, local wind-induced vertical mixing seems to have amplified the impact of the inflowing $\mathrm{CO_2}$-enriched water mass on the $p\mathrm{CO_2}$ in the surface water by increasing its input into surface layers. Accordingly, the $\mathrm{CO_2}$-enriched waters were apparently supplied from somewhere else. Since the NECC carries offshore waters towards the Costa Rican shore during the non-upwelling season (Wyrtki, 1965, 1966; Fiedler, 2002), it is assumed that the $\mathrm{CO_2}$-enriched subsurface water originated

somewhere south of our study area in the open ETP. The absence of such a cold-event during the non-upwelling season in 2013 shows that it is an irregular feature (Fig. 2c, d), whereas the driving forces are still elusive. Nevertheless, it affects the metabolic processes in the bay as will be discussed in the following section, which analyzes the daily cycles during the non-upwelling seasons in 2012 and 2013.

**4.2 Processes behind the variability of the carbonate system**

In 2012, the pH and the $p\mathrm{CO_2}$ values followed a pronounced diurnal cycle with highest pH and lowest $p\mathrm{CO_2}$ values during the late afternoon and lowest pH and highest $p\mathrm{CO_2}$ values around sunrise in the early morning (Fig. 4a). Such daily cycles are typical for tropical regions and are assumed to be caused by photosynthesis during the day and respiration of organic matter during the night (Shaw et al., 2012; Albright et al., 2013; Cyronak et al., 2013a). The aragonite saturation state as well as the DIC/TA ratio followed this pattern, with higher $\Omega_a$ and lower DIC/TA ratio values during the day as well as lower $\Omega_a$

and higher DIC/TA values at night (Fig. 4b).

In contrast to 2012, the pH in 2013 hardly followed any daily cycle, while the $p\mathrm{CO_2}$ cycles were similar in both periods (Fig. 4). To characterize the relative importance of the processes responsible for the observed changes in pH and $p\mathrm{CO_2}$ (photosynthesis, respiration, calcification and dissolution) we used the model described earlier, which is based on the determined DIC concentrations during times when pH and $p\mathrm{CO_2}$ revealed their daily minima and maxima, respectively. For

example, if photosynthesis of organic matter dominates the transition from early morning maxima of $p\mathrm{CO_2}$ to late afternoon minima of $p\mathrm{CO_2}$ it should be associated with a loss of DIC. Whether photosynthesis was accompanied with enhanced calcification can be detected by an associated decrease of TA. Since decreasing DIC raises the pH and a decrease in TA lowers the pH, such photosynthetic enhanced calcification hardly affects the pH and could explain the weak daily cycle



observed in 2013. Alternatively, if photosynthesis is accompanied by carbonate dissolution during the day, this would amplify the daily cycle of pH and $p\text{CO}_2$ as seen during the cold-water intrusion event in 2012.

These daily cycles in pH and $p\text{CO}_2$ suggest concordant cycles in DIC and TA, which unexpectedly could not be observed. The interplay of all the four metabolic processes of relevance (photosynthesis, respiration, calcification and dissolution)

seem to mask pronounced daily cycles of DIC and TA, but the pronounced daily cycle of the DIC/TA ratio indicate that some of these processes control the daily cycle of pH and $p\text{CO}_2$ (Fig. 4).

To identify the dominant processes, we developed a numerical model and recalculated the daily trends of pH and $p\text{CO}_2$ (Fig. 5). The calculated $p\text{CO}_2$ and pH agree quite well to the measured ones and support our previous interpretations that during 2013, photosynthesis and light-enhanced calcification prevailed during the day and carbonate dissolution was relegated to

night hours alongside respiration. During the 2012 cold-water intrusion event, the system was dominated by photosynthesis and calcium carbonate dissolution during light hours, with respiration and calcification occurring at night. Dissolution taking place during daytime is particular but not completely unusual, as it has been reported on tropical sandy bottoms under ambient (Yates and Halley, 2006; Cyronak et al., 2013b) and high-$\text{CO}_2$ conditions (Comeau et al., 2015). Dark-calcification is not entirely uncommon and occurs in both, sandy bottoms and coral reefs (Yates and Halley, 2006b; Albright et al., 2013).

Accordingly, the entrainment of $\text{CO}_2$-enriched water from the NECC seems to shift the carbonate chemistry of Bahía Culebra from a system where photosynthesis and calcification are the controlling processes during light hours to a system in which daytime is dominated by photosynthesis and dissolution. The net effect, as observed, is an enhanced $p\text{CO}_2$ and lower $\Omega_a$ during periods characterized by the inflow of $\text{CO}_2$-enriched waters (Table 1). This has strong ecological implications for local coral reef ecosystems.

**4.3 Ecological implications for coral reefs**

Coral reefs in Bahía Culebra were dominated by *Pocillopora* spp. and *Pavona clavus* (Jiménez, 2001; Jiménez et al., 2010), whereas *Porites lobata* is the main reef forming coral in the southern part of the Costa Rican Pacific coast (Cortés and Jiménez, 2003; Glynn et al., 2017). Although the reefs in the north are naturally exposed to periodic high-$\text{CO}_2$ conditions during upwelling events (Rixen et al., 2012), as well as during cold water intrusions in non-upwelling season, the linear

extension rates of *Pocillopora* spp. and *P. clavus* exceeded those of the same species in other regions (Fig. 6) (Glynn, 1977; Jiménez and Cortés, 2003; Manzello, 2010a; Rixen et al., 2012). This suggests that local corals are adapted to the upwelling of cold and acidic waters.

Since it is generally assumed that $\Omega_a$ strongly influences coral growth, we calculated the annual mean $\Omega_a$ of 3.04 for Bahía Culebra by integrating measurements from all periods (Table 1). This average $\Omega_a$ correlates well with the extension rates of

*Pocillopora damicornis* and *P. clavus* from three upwelling-influenced areas in the ETP and supports the general assumption that $\Omega_a$ controls the growth of corals (Fig. 6).



The dependency of coral growth on $\Omega_a$ and the mean $\Omega_a$ (2.71) during the upwelling season (Table 1) suggests that upwelling of acidic waters should reduce the linear extension of these corals during upwelling season. The increased $\Omega_a$ during non-upwelling season in turn must enhance linear extension and explains corals' high annual mean growth rates. This furthermore implies the sensitivity of coral growth within the bay to the occurrence of cold events that lower $\Omega_a$ during the

non-upwelling season, as seen in June 2012. The $\Omega_a$ values from this study suggest that most favorable conditions for coral growth occur during non-upwelling season, period that coincides with development of the rainy season. This implies that during the main growing season the eutrophication and siltation caused by human impacts on river discharges, as well as the development of harmful algal blooms, could also strongly affect the corals' annual mean growth rates (Cortés and Reyes-Bonilla, 2017).

Despite the corals' high annual mean linear extension rates, net accretion rates of the reefs in Bahía Culebra are relatively low (Alvarado et al., 2012) and reef frameworks of *Pocillopora* spp. hardly exceeded a thickness of 0.5 m during the period of our observation. This denotes that although *Pocillopora* spp. and *P. clavus* are adapted to the entrainment of acidic waters, these reefs are growing on the verge of their ecological tolerance because of bioerosion and reef erosion due to physical forces. Corals' accretional gaps, which are known from the geological record, document the vulnerability of these reefs.

They have been linked to increased ENSO variability (Toth et al., 2012, 2015) and stronger upwelling conditions (Glynn et al., 1983), favoring dissolution and erosion of reef frameworks while at the same time restricting coral growth.

The y-intercept of the regression equation derived from the correlation between linear extension and $\Omega_a$ furthermore implies that linear extension of *P. damicornis* and *P. clavus* approaches zero under a carbonate saturation state of $\Omega_a < 2.5$ (*P. damicornis*) and $< 2.2$ (*P. clavus*). According to climate predictions, the global $\Omega_a$ will reach values $< 2.0$ by the end of this

century (IPCC, 2014), suggesting that OA seriously threatens the reefs in Bahía Culebra. This emphasizes the importance of the Paris agreement and all the global efforts to reduce the $CO_2$ emission into the atmosphere (Figueres et al., 2017).

## 5 Conclusions

This study builds on previous field studies in the upwelling areas of Panamá (Manzello et al., 2008; Manzello, 2010b) and Papagayo (Rixen et al., 2012), contributing with data from in situ measurements carried out within a system that is naturally

exposed to low-pH conditions. The results presented suggest that coral reefs from Bahía Culebra are exposed to a high intra- and interannual variability in the carbonate system. Challenging conditions are not restricted to the upwelling season, they occur sporadically also during non-upwelling seasons, when pH and $CO_2$ concentrations reach values comparable to those during upwelling events. Linear extension rates of the main reef building corals in the bay are sensitive to changes in $\Omega_a$, suggesting that upwelling reduces coral growth by introducing acidic subsurface waters in the surface layers. Rising levels of

$\Omega_a$ enhance coral growth during the non-upwelling season due to which the linear extension rates of the main reef building corals in Bahía Culebra were among the highest in the ETP; however, reef accretions was low due to erosion. The latter indicates a sensitivity of coral reefs to the intensity of the upwelling and other processes occurring during the non-upwelling/rainy season, such as human impacts on river discharges, the occurrence of cold events (e.g. 2012), and ultimately





to OA. Threshold values of $\Omega_a$ when coral growth likely approaches zero were derived from the correlation of $\Omega_a$ and linear extension rates and this suggests that OA will seriously threat reefs in Bahía Culebra, which are already at the verge of their ecological tolerance.

## 6 Data availability

Data are available by direct request to the corresponding author.

## 7 Author contribution

C. Sánchez-Noguera designed the study, collected and analyzed the samples, analyzed the data, prepared figures and/or tables and wrote the paper. I. Stuhldreier, J. Cortés, Á. Morales, C. Jiménez and C. Wild reviewed the paper. T. Rixen designed the study and review the paper.

## 8 Competing interests

The authors declare that they have no conflict of interest.

## 9 Disclaimer

This study was funded by the Leibniz Association, as part of the PhD research of C. Sánchez-Noguera. Funders had no role in conceiving the study, collection and analysis of data or manuscript preparation.

## 10 Acknowledgements

This project was conducted in cooperation with the Centro de Investigación en Ciencias del Mar y Limnología (CIMAR), University of Costa Rica. Special thanks to Marina Papagayo for allowing us to deploy the sensors in their facilities, Giovanni Bassey and Carlos Marenco for logistic support and sample collection.

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





**Table 1: Measured and calculated (\*) parameters, during upwelling (2009) and non-upwelling seasons (2012, 2013) at Bahía Culebra, Costa Rica.**

|  | pH (Total scale) | $p$CO$_2$ (μatm) | CO$_2$ (μmol kg$^{-1}$) | T (°C) | DIC* (μmol kg$^{-1}$) | TA* (μmol kg$^{-1}$) | Ω* |
|---|---|---|---|---|---|---|---|
| **2009** | | | | | | | |
| Mean ± SD | 7.91 ± 0.32 | 578.49 ± 42.82 | 16.44 ± 1.35 | 25.09 ± 0.57 | 2098.71 ± 103.81 | 2328.42 ± 118.45 | 2.71 ± 0.29 |
| Range | 7.84-7.99 | 474.43-643.52 | 13.32-18.51 | 23.88-26.10 | 1610.34-2360.92 | 1787.57-2610.47 | 1.92-3.35 |
| **2012** | | | | | | | |
| Mean ± SD | 7.98 ± 0.04 | 456.38 ± 69.68 | 11.77 ± 1.99 | 29.61 ± 0.93 | 1924.65 ± 195.07 | 2204.54 ± 212.18 | 3.32 ± 0.46 |
| Range | 7.81-8.06 | 302.32-658.32 | 7.64-17.83 | 27.13-31.37 | 1476.90-2355.39 | 1715.00-2714.84 | 1.89-4.84 |
| **2013** | | | | | | | |
| Mean ± SD | 8.02 ± 0.03 | 375.67 ± 24.25 | 9.56 ± 0.64 | 30.08 ± 0.27 | 1800.92 ± 142.78 | 2102.66 ± 174.79 | 3.50 ± 0.49 |
| Range | 7.95-8.10 | 325.40-447.45 | 8.22-11.52 | 29.31-30.72 | 1412.07- 2199.50 | 1637.76-2599.99 | 2.36- 4.92 |



**Table 2: Correlations between tide height and four parameters during non-upwelling season (2012, 2013).**

| Year | pH | $p\mathrm{CO_2}$ | T | Wind |
|------|------|------|------|------|
| 2012 | -0.004 | 0.037 | -0.005 | 0.033 |
| 2013 | 0.111 | 0.026 | -0.093 | -0.126 |

All p-values > 0.05

25

30



**Figure 1: Location of Bahía Culebra (square) in the Gulf of Papagayo, North Pacific coast of Costa Rica (insert). Measurements were made at Marina Papagayo (star). Main ocean currents influencing the Gulf of Papagayo (dashed arrows): NECC= North Equatorial Counter Current, CRCC= Costa Rica Coastal Current.**





**Figure 2: Measured parameters (wind speed, SWT, pH and $p$CO₂) during the non-upwelling seasons of June 2012 (a, b) and May-June 2013 (c, d), at Bahía Culebra. Shaded area in (a) and (b) indicates the 2012 upwelling-like event.**



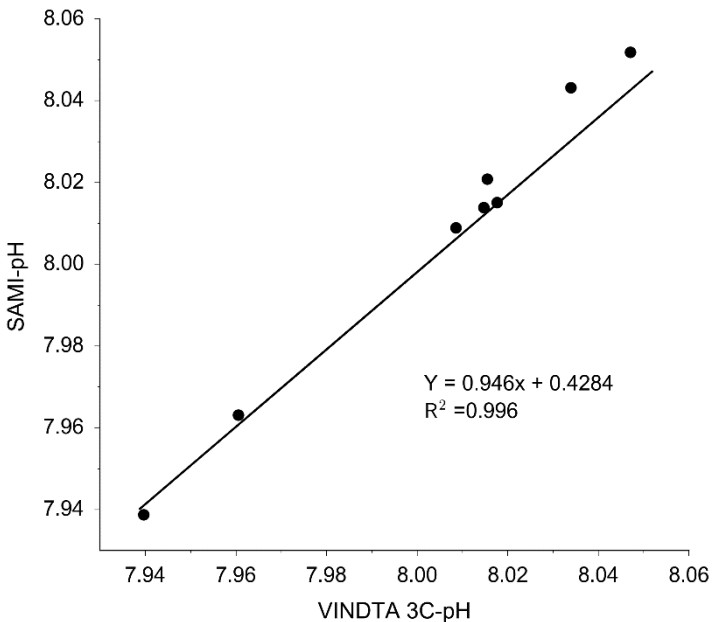

**Figure 3: Validation of SAMI sensors using discrete samples measured with a VINDTA 3C system.**





**Figure 4: Diel pattern of parameters measured in Bahía Culebra. Data points are hourly averages of 15 and 7 consecutive days in 2012 (a, b) and 2013 (c, d), respectively. The shaded area represents daylight hours.**



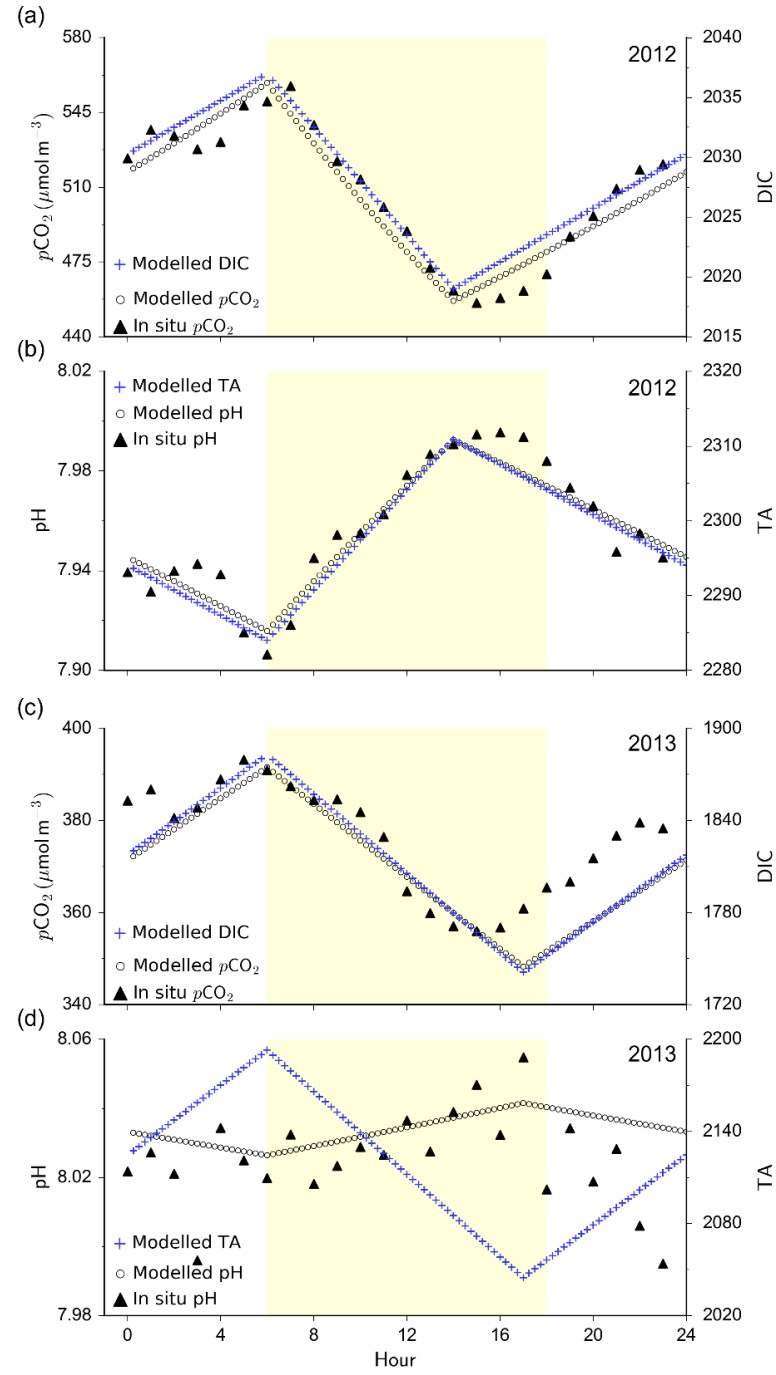

**Figure 5: Expected diel behavior of the carbonate system in 2012 (a, b) and 2013 (c, d), based on measured parameters. Modeled parameters are shown as blue crosses and empty circles, the reference parameter used to adjust the model is shown in black triangles. Shaded area represents daylight hours.**



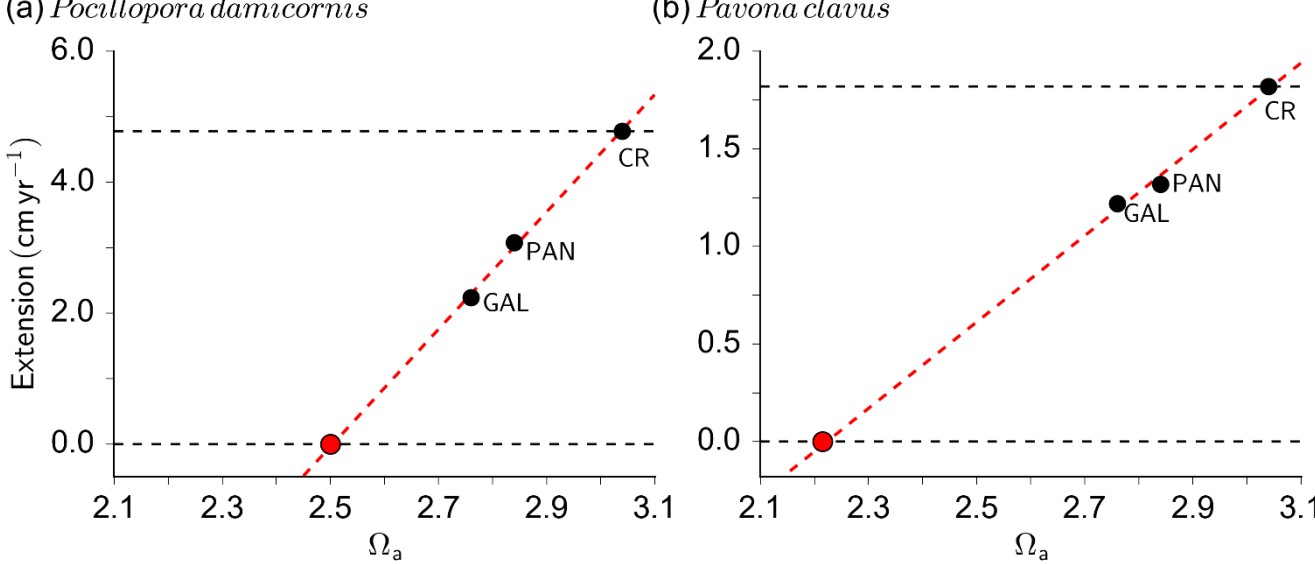

**Figure 6: Mean aragonite saturation state ($\Omega_a$) versus mean linear extension rates of *Pocillopora damicornis* (a) and *Pavona clavus* (b) from upwelling areas in Costa Rica (CR), Panamá (PAN) and Galápagos (GAL) (Jiménez and Cortés, 2003; Manzello, 2010a). Red broken line shows the regression equation estimated by Rixen et al. (2012), the red mark represents our estimated $\Omega$ for Bahía Culebra when coral growth equals zero.**