# Peer review of "Natural ocean acidification at Papagayo upwelling system (North"

_Biogeosciences, 2017_

## Referee Comment (RC1) · Anonymous Referee #1 · 1 Dec 2017

General comments: This manuscript describes the intra-annual variability in carbonate chemistry in a naturally acidified environment in Pacific Costa Rica. As the authors point out, such natural laboratories are very important to improving our understanding of coral-reef development under future climate change. The manuscript is generally clear and well-written and the implications of their findings to the future of coral growth and reef accretion are significant. It is, therefore, my suggestion that the manuscript be accepted to Biogeosciences after some, generally minor, revisions, as described below:

First, I think it's important that the authors make it clear at the beginning of the

manuscript (Abstract and Methods) that they did not measure coral growth in this study that they used previously published data to evaluate the relationships between carbonate chemistry and coral growth, because this was not obvious to me until the Discussion. I would also say what studies you derived the ecological data from in the Methods and include a brief description of the methodologies those researchers used so that readers can evaluate those data.

The authors should also consider adding some additional text in the Methods/Discussion regarding the collection and interpretation of the carbonate-chemistry data. I've listed a number of specific comments below, but my most significant concern is with the statement that DIC is temperature-independent. Temperature impacts the solubility of CO2 and directly impacts DIC. Additionally. since the DIC data are derived from pCO2 measurements, which are dependent on T, this does not get rid of the temperature effect on pCO2. I would like to see the author address this issue more thoroughly.

Specific comments:

P2, L25: But see Toth et al. 2012. Accretion rates in Panama when reefs were growing were comparable to rates observed on reefs in the Caribbean and the authors did not find a significant difference in reef accretion between upwelling and non-upwelling sites. I would suggest making the language here more conservative.

P3, L23: Please provide information on the pH scale used as suggested by the OA best practices: http://oceanrep.geomar.de/8471/1/Guide%20best%20practices%202011.pdf

P3, L28: How can you be sure that the carbonate chemistry at 30 cm was the same as where the instrument was (1.5 m)?

P4, L8: Did Rixen et al. 2012 use the same methodology for measuring carbonate chemistry? I think it's important to include this information so that readers can fully

evaluate the results.

P4, L11-12: I would suggest including the statement "All GLM assumptions were met" here instead of in the Results.

P4, L19: Please include a citation for the rain-ratio (Archer and Maier-Reimer 1994?) and describe what it is

P4, L23: What are the ROI values of -2.6 and 0.8 based on. Please explain in the text where these numbers came from.

P5, L16-17: Temperature variability also can impact DIC. Also since these data are derived from pCO2 measurements, which are dependent on T, this does not get rid of the temperature effect. I would like to see the author address this issue more throughly.

P5, L22: include error term for the aragonite saturation state

P6, L5-7: It would be more informative to talk about how much DIC dropped during the "upwelling-like" event in 2012 than to talk about the overall average for the whole sampling period.

P6, L16: You can't really make conclusions about how regular these sorts of events are based on a few weeks of data from just two years. I would change the wording here to reflect this.

P6, L26: The wording of this sentence is confusing. I would rephrase to something like: Although the pCO2 cycles in 2013 followed a similar pattern to 2012, pH was more variable (or less predicable). I would also move this sentence to the end of the previous paragraph and start the next paragraph with "To characterize..."

P7, L3: I don't understand what this sentence means. Which cycles? Could not be observed in the data? In the model? I actually think that this whole paragraph needs to be fleshed out more. It's not clear to me what the authors are trying to say.

P7, L10: add a reference to Fig. 5c&d at the end of the sentence ending with "respiration"

P7, L10: Clarify whether you are describing changes that just happened during the cold-water event or during the entire sampling period in 2012.

P7, L30-31: This sentence is misleading. I think that what you mean to say is that saturation state in each of those locations predicts extension rate. I'm assuming that you used the saturation states/linear extension determined by Manzello 2010 for Panama and Galapagos, but this is not clear in either the text or the figure.

P8, L11-12: I think it would be good to compare your observation of reef thickness with reef thickness elsewhere in Costa Rica or elsewhere in the ETP, as most reader won't be familiar with how thick a reef "should" be see:

Glynn, P. W., E. M. Druffel, and R. B. Dunbar. 1983. A dead Central American coral reef tract: Possible link with the Little Ice Age. Journal of Marine Research 41:605–637.

Cortés, J., I. G. Macintyre, and P. W. Glynn. 1994. Holocene growth history of an eastern Pacific fringing reef, Punta Islotes, Costa Rica. Coral Reefs 13:65-73.

Toth et al. 2012, Science

or summary in: Toth, L.T., I.G. Macintyre, and R.B. Aronson. 2017. Holocene reef development in the Tropical Eastern Pacific. In Glynn, P.W., D.P. Manzello, and I.C. Enochs (eds). Coral Reefs of the Eastern Pacific: Persistence and Loss in a Dynamic Environment. Springer-Verlag, New York. doi: 10.1007/978-94-017-7499-4_6

P8, L14: It's not correct to talk about "coral" accretion because accretion is a net measure of the growth of a reef. I would re-phrase the beginning of this sentence to something like: Gaps in coral-reef accretion. I would also specify that these gaps are known from the geological record of the ETP. They are not common elsewhere.

P8, L20: It's likely that OA will threaten places like Bahia Culebra sooner than elsewhere right? It might be interesting to include a brief discussion of when/how OA may

impact reefs in the ETP vs elsewhere and what the implications of this would be.

P8, L31: I would make it clear here that previous studies have shown that reef accretion is low in Bahia Culebra.

Technical corrections:

P1, L18: change to: 2009 upwelling season or upwelling season in 2009

P1, L22: change to: also occurs sporadically in the non-upwelling season

P1, L25: change to: during the upwelling season fall

P1, L28: I would add a phrase like the continued growth of between threaten and reefs

P2, L5: Delete "ones"

P2, L6: Add comma after references

P2, L14: Delete extra "A"

P3, L25: add was before "used"

P4, L20: add hyphen between "rain" and "ratio"

P4, L21-22: change / to and to be consistent with the ends of the sentences

P4, L21: comma before "respectively"

P5, L9: change "did" to was

P5, L15: change "between" to among

P5, L30: delete "an"

P6, L2: change "those" to that

P6, L4: change "to" to with

P6, L6: change "fell below" to were lower than

P6, L9: I would suggest starting a new paragraph here

P6, L9: delete "only" change to: had already increased by June 7th

P6, L13: Somewhere else than where? maybe just change to something like: "a different location than they are during the upwelling season"

P6, L16: change "whereas" to and. I would also suggest rewording "it affects the" to something like: these types of events have the potential to affect the...to be more conservative

P6, L17-18: I don't think the phrase at the end of this sentence is necessary

P6, L31: change "loss of" to decline in

P7, L12: I don't understand the phrase "is particular but not completely unusual". Do you mean peculiar?

P7, L13: I suggest changing the beginning of the sentence here to: Similarly, dark...

P8, L3: change the end of this sentence to read: corals' relatively high annual mean growth rates in Bahia Culebra.

P8, L4: I would change the beginning of this sentence to: implies that coral growth should be sensitive...

P8, L6: add the before "period"

P8, L18: I suggest saying should approach

P8, L24: I suggest changing "contributing with" to that provide and "carried out with" to from

P8, L28-30: reef-building corals needs a hyphen

P8, L30: change "due to which" to which may explain why

P8, L31: delete "s" at end of accretion

---

## Referee Comment (RC2) · Anonymous Referee #2 · 12 Dec 2017

Overall

The authors present interesting time-series data of pH and pCO2 from an upwelling region of Pacific Costa Rica. These data are much needed and this contributes to our limited understanding of CO2 dynamics in the eastern Pacific. I do believe there is a publication nested within this draft, but there will need to be a significant cutting down and revision based on my comments below. The paper will be considerably shortened and changed slightly in scope, but still is publishable.

Big comments

1)The calculation of TA and DIC from pH and pCO2 is unreliable and prone to large

errors (see Cullison-Gray et al. 2011; Gray et al. 2011, which you cite). This is probably why you are reporting such unbelievably high values of TA (>2600) and DIC (>2300). Not only do these values never occur on coral reefs (to my knowledge), even in strong upwelling zones like the ETP, you are also reporting relatively low salinities of 32.5. This makes these high values all the more suspect because if you calculate salinity normalized for a TA = 2715, you get NTA= 2923! These types of DIC and TA values are unheard of from reef environments.

As such, you need to remove all DIC and TA values that were calculated from pH and pCO2, including the model you generated from some of those values to interpret rates of metabolism driving diurnal changes. Unfortunately, this all has to be deleted from the paper.

2) Please be clear as to what you actually measured versus what you hypothesize, or think is going on. For instance, in the abstract you discuss how calcification is enhanced during the non-upwelling season when saturation states are elevated. This may be true, but at this point it is a hypothesis as you have no data to support it. There are several other instances where hypotheses are discussed as fact, i.e., reef thickness and accretion etc.

3) What is the proximity of the instruments to an actual coral reef? You need to be clear about this because what you are measuring may actually be a result of the metabolism of the carbonate benthos directly under the dock plus the water column processes rather than anything to do with reef dynamics.

4) Finally, the paper is generally well written, but would benefit from being proofread by a native English speaker.

Specific comments

1) P2, line 19. You can't say adaptation without genetics work. Reword to say "adaptation or acclimatization"

2) P2, last line. Thus far there is really no data out there suggesting that OA will directly lead to coral mortality. In fact, in a seminal study, Fine and Tchernov (2007, Science) exposed two coral species to undersaturation and found they stopped growing skeletons and lived in an anemone-like state. Once conditions were raised above saturation, they began producing skeletons again.

3) P3, section 2.2. You discuss taking discrete samples, but do not say how many or how often. Please mention this. The values of TA >2600 and DIC > 2300 have to be calculated from pH-pCO2. You show the result of 8 bottle samples pH relative to the SAMI pH – what about the SAMI CO2 or any other data calculated from the bottles?

4) P5, salinity is a unitless value. PSU describes the scale and is not a unit

5) P5, line 5. Need to say what the number after the plus and minus is the first time you use it.

6) P7, 1st line – why couldn't stimulation of photosynthesis from higher nutrients during upwelling be causing the large amplitude? You can't speak to dissolution without actual TA data and increased photosynthesis seems more likely.

7) P7, line 25. Say adapted and or acclimatized.

8) P7, last line. Reword to say saturation sate is one of the controlling factors in coral growth. Its likely of secondary or tertiary level importance behind temperature and light.

9) P8, delete sentence spanning lines 3 to 5 as well as last sentence of 1st paragraph

10) P8, line 10. Did you measure reef thickness? Also, I'd avoid saying they are growing on the edge of their ecological tolerance. Its not clear what is meant by that.

11) P8, last 5 or so sentences, a lot of conjecture written as fact.

12) Figure 4, your units for pCO2 are incorrect

---

## Author Comment (AC1) · 29 Jan 2018

We appreciate all comments and suggestions from anonymous referees. In the following paragraphs we listed our response to the comments from Anonymous Referee #1:

[RC1] General comments: This manuscript describes the intra-annual variability in carbonate chemistry in a naturally acidified environment in Pacific Costa Rica. As the authors point out, such natural laboratories are very important to improving our understanding of coral-reef development under future climate change. The manuscript is generally clear and well-written and the implications of their findings to the future of

coral growth and reef accretion are significant. It is, therefore, my suggestion that the manuscript be accepted to Biogeosciences after some, generally minor, revisions, as described below:

First, I think it's important that the authors make it clear at the beginning of the manuscript (Abstract and Methods) that they did not measure coral growth in this study that they used previously published data to evaluate the relationships between carbonate chemistry and coral growth, because this was not obvious to me until the Discussion. I would also say what studies you derived the ecological data from in the Methods and include a brief description of the methodologies those researchers used so that readers can evaluate those data.

REPLY: We will make this point clear in the first sections of the manuscript (Abstract and Methods). Also will include in the Methods section the corresponding references of studies from which we took the coral growth rates.

[RC1] The authors should also consider adding some additional text in the Methods/Discussion regarding the collection and interpretation of the carbonate-chemistry data. I've listed a number of specific comments below, but my most significant concern is with the statement that DIC is temperature-independent. Temperature impacts the solubility of $CO_2$ and directly impacts DIC. Additionally, since the DIC data are derived from $pCO_2$ measurements, which are dependent on T, this does not get rid of the temperature effect on $pCO_2$. I would like to see the author address this issue more thoroughly.

REPLY: What we wanted to estimated was to which extend the observed changes of $pCO_2$ were caused by changes in temperature and/or DIC concentrations. We will express this more clearly.

Specific comments:

[RC1] P2, L25: But see Toth et al. 2012. Accretion rates in Panama when reefs

were growing were comparable to rates observed on reefs in the Caribbean and the authors did not find a significant difference in reef accretion between upwelling and non-upwelling sites. I would suggest making the language here more conservative.

REPLY: We will modify the sentence as follow:

Original: "...resulting in poorly cemented coral reefs with low accretion rates that are subject to rapid bioerosion"

Modified: "...and have the potential to produce poorly cemented coral reefs with low accretion rates that are subject to rapid bioerosion"

[RC1] P3, L23: Please provide information on the pH scale used as suggested by the OA best practices: http://oceanrep.geomar.de/8471/1/Guide%20best%20practices%202011.pdf

REPLY The pH scale will be included => total hydrogen ion scale

[RC1] P3, L28: How can you be sure that the carbonate chemistry at 30 cm was the same as where the instrument was (1.5 m)?

REPLY: We can't. But the samples were taken 30 cm below the surface, which is 1.2 m above the instrument and up to 7.7 m above the bottom (the distance above bottom varied with tide cycles). Since we have never seen any indication of depth related changes (e.g. in turbidity) we assume that DIC and TA changes within the water column are negligible.

[RC1] P4, L8: Did Rixen et al. 2012 use the same methodology for measuring carbonate chemistry? I think it's important to include this information so that readers can fully evaluate the results.

REPLY: The following sentence will be included: "In 2009 xCO2 was measured by an underway pCO2 system (SUNDANS) equipped with an infrared gas analyzer (LI-7000), and pH was measured using an Orion ROSS electrode and an Orion StarTM."

[RC1] P4, L11-12: I would suggest including the statement "All GLM assumptions were met" here instead of in the Results.

REPLY: Will be included as follow: "The GLM was evaluated using graphical methods to identify violations of assumptions of homogeneity of variance and normality of residuals. All GLM assumptions were met."

[RC1] P4, L19: Please include a citation for the rain-ratio (Archer and Maier-Reimer 1994?) and describe what it is

REPLY: The term rain-ratio seems to be confusing since it actually describes the ratio between the export of organic carbon and calcium carbonate carbon. Accordingly, it will be changed as follows: "ROI describes the ratio between the production of organic carbon (POC) and precipitation of calcium carbonate carbon (PIC) (Archer and Maier-Reimer, 1994), and was used to link $\Delta$POC to $\Delta$PIC (ROI=POC/PIC) (Eq. 2, 3)."

[RC1] P4, L23: What are the ROI values of -2.6 and 0.8 based on. Please explain in the text where these numbers came from.

REPLY: These numbers were used to run the model in order to obtain the best fit between measured and calculated pH and pCO2 values. The assumption of calcium carbonate dissolution caused the negative sign.

Modified sentence will read as follow: "To verify the results from the model, we used the output $\Delta$DIC and $\Delta$TA to calculate new pCO2 and pH values, which were further compared to the measured ones. The best fit between modeled and measured values was achieved with a respective ROI of -2.6 for 2012 and 1.0 for 2013, whereas the assumption of calcium carbonate dissolution caused the negative sign."

[RC1] P5, L16-17: Temperature variability also can impact DIC. Also since these data are derived from pCO2 measurements, which are dependent on T, this does not get rid of the temperature effect. I would like to see the author address this issue more throughly.

REPLY: What we meant to say was that we tried to estimate to which extends temperature and/or changes in the DIC concentrations had an effect on the observedvariations of pCO2. This will be clarified.

[RC1] P5, L22: include error term for the aragonite saturation state

REPLY: Standard deviation (SD) will be included as follows: $3.06 \pm 0.49$

[RC1] P6, L5-7: It would be more informative to talk about how much DIC dropped during the "upwelling-like" event in 2012 than to talk about the overall average for the whole sampling period.

REPLY: DIC, TA and pCO2 increased during the "upwelling-like" event. The only parameter that decreased during this period was pH. We will include one sentence describing in more detail the behavior of pH (in a similar way as how was done for pCO2). "Similarly, this increase in pCO2 was accompanied by a dropped in pH from 8.04 to 7.83"

[RC1] P6, L16: You can't really make conclusions about how regular these sorts of events are based on a few weeks of data from just two years. I would change the wording here to reflect this.

REPLY: We propose the following modification: Original: "The absence of such a cold-event during the non-upwelling season in 2013 shows that it is an irregular feature…"

Modified: "The absence of such a cold-event during the non-upwelling season in 2013 suggests that the occurrence of this kind of events might be an irregular feature…"

[RC1] P6, L26: The wording of this sentence is confusing. I would rephrase to something like: Although the pCO2 cycles in 2013 followed a similar pattern to 2012, pH was more variable (or less predicable). I would also move this sentence to the end of the previous paragraph and start the next paragraph with "To characterize..."

REPLY: 1) It will read as follows: "Although the pCO2 cycles in 2013 followed a similar

pattern to 2012, pH cycles were less predictable." 2) Sentence will be moved to the end of the previous paragraph and the next paragraph will start with "To characterize..."

[RC1] P7, L3: I don't understand what this sentence means. Which cycles? Could not be observed in the data? In the model? I actually think that this whole paragraph needs to be fleshed out more. It's not clear to me what the authors are trying to say.

REPLY: The first sentence is certainly confusing, and after a careful reading we concluded that it has no relevance in transmitting the take-home message, which basically is: pH and pCO2 show pronounced daily cycles but the interaction of metabolic processes difficult to identify a clear pattern in the daily cycles of DIC and TA.

To improve reader's understanding this first sentence of that paragraph will be removed and the remaining sentence will be modified and merged with the next paragraph. After these changes the resulting paragraph will read as follows:

"The interplay of all the metabolic processes (photosynthesis, respiration, calcification and dissolution) seems to be softening the daily cycles of DIC and TA. However, the daily patterns of the DIC/TA ratio raise some clues about the processes that are controlling the daily cycles of pH and pCO2 (Fig. 4). To identify the dominant processes, we developed a numerical model and recalculated the daily trends of pH and pCO2 (Fig. 5)."

[RC1] P7, L10: add a reference to Fig. 5c&d at the end of the sentence ending with "respiration"

REPLY: It will read as follows: "...during the day and carbonate dissolution was relegated to night hours alongside respiration (Fig. 5c, d)".

[RC1] P7, L10: Clarify whether you are describing changes that just happened during the cold-water event or during the entire sampling period in 2012.

REPLY: We are describing the changes during the cold-water event (7 days), not during the full period 2012 (15 days). For a better understanding, the sentence will be modified

in the following way: "Based on the observed daily pH and pCO2 cycles and in line with our model results, it seems that during the 2012 cold-water intrusion event (June10-17), the diurnal cycle of the carbonate system was dominated by photosynthesis and..."

[RC1] P7, L30-31: This sentence is misleading. I think that what you mean to say is that saturation state in each of those locations predicts extension rate. I'm assuming that you used the saturation states/linear extension determined by Manzello 2010 for Panama and Galapagos, but this is not clear in either the text or the figure.

REPLY: The work by Manzello was cited in the text and in Fig. 6. However, we will modify it to make it clearer that we used those values as input for our figure.

Writing in text will be improved in the following way:

"Aragonite saturation state ($\Omega$a) is known as one of the main variables influencing coral growth and therefore reef distribution around the world (Kleypas et al. 1999). By integrating the data from the present study and values previously reported by Rixen et al. (2012), we estimated that the annual mean $\Omega$a in Bahía Culebra is 3.06. Additionally, earlier studies in the ETP measured $\Omega$a values and coral extension rates from locations that are under the influence of upwelling events (Manzello 2010a), whilst extension rates from Bahía Culebra were measured by Jiménez and Cortés (2003). The correlation between our estimated $\Omega$a with the available data from Bahía Culebra, Panamá and Galápagos indicates that coral extension rates in each of those locations are predicted by their corresponding $\Omega$a values (Fig. 6)."

Figure caption will be modified as follow: "Mean aragonite saturation states ($\Omega$a) - from present and former studies - versus previously reported mean linear extension rates of Pocillopora damicornis and Pavona clavus from upwelling areas in Costa Rica (CR) (Jiménez and Cortés, 2003), Panamá (PAN) and Galápagos (GAL) (Manzello, 2010a). Red broken line shows the regression equation as estimated by Rixen et al. (2012). Red mark represents our estimated $\Omega$a threshold for Bahía Culebra, when coral growth equals zero"

[Figure]

[RC1] P8, L11-12: I think it would be good to compare your observation of reef thickness with reef thickness elsewhere in Costa Rica or elsewhere in the ETP, as most reader won't be familiar with how thick a reef "should" be see:

REPLY: P8 L10-11 will be modified as follows: "Despite the corals' high annual mean linear extension rates, during the period of our observation the reef frameworks of Pocillopora spp in Bahía Culebra hardly exceeded a thickness of 0.5 m."

The following sentence will be included in P8, L12:

"In the Pacific coast of Costa Rica, the maximum levels of Holocene framework accumulation range from < 3 to 9 m in Pocillopora-dominated reefs and from 3 m to 11 m in reefs built by massive species, with thinner frameworks in upwelling-influenced locations (Toth et al., 2017)."

[RC1] P8, L14: It's not correct to talk about "coral" accretion because accretion is a net measure of the growth of a reef. I would re-phrase the beginning of this sentence to something like: Gaps in coral-reef accretion. I would also specify that these gaps are known from the geological record of the ETP. They are not common elsewhere.

REPLY: Our sentence will be modified:

Original: "Coral's accretional gaps, which are known from the geological records, document the vulnerability of these reefs. They have been..."

Modified: "Gaps in coral reef accretion at the ETP are known from the geological record (Toth et al. 2017). They have been..."

[RC1] P8, L20: It's likely that OA will threaten places like Bahia Culebra sooner than elsewhere right? It might be interesting to include a brief discussion of when/how OA may impact reefs in the ETP vs elsewhere and what the implications of this would be.

REPLY: We will include the following sentences just before the last sentence of the current paragraph:

"Worldwide, OA is expected to reduce coral reefs' resilience by decreasing calcification and increasing dissolution and bioerosion (Kleypas et al., 1999; Yates and Halley, 2006a; Anthony et al., 2011). Coral reefs from the ETP are affected by chronic and acute disturbances, such as thermal stress and natural acidification resulting from ENSO and upwelling events, respectively (Manzello et al., 2008; Manzello, 2010b). Historically, these reefs have shown a high resilience to both stressors by separately but their coupled interaction can cause coral reef lost within the next decades. The ETP have the lowest $\Omega$a of the tropics, near to the threshold values for coral reef distribution, therefore the reefs from this region may be the most affected by the increasing levels of anthropogenic CO2 and also show the first negative impacts of this human induced OA (Manzello et al., 2017)."

[RC1] P8, L31: I would make it clear here that previous studies have shown that reef accretion is low in Bahia Culebra.

REPLY: We have modified the sentence in section 4.3 (P8, L10-11) to clarify this point; therefore we consider that this statement can be removed from the "Conclusions" section. Sentences spanning lines P8, L28 to P9, L1 will be modified as follow:

Original sentence: "Rising levels of $\Omega$a enhance coral growth during the non-upwelling season due to which the linear extension rates of the main reef building corals in Bahía Culebra were among the highest in the ETP; however, reef accretion was low due to erosion".

Modified sentence: "Previous studies reported that the linear extension rates measured in Bahía Culebra were among the highest in the ETP, thus is likely that coral growth in this bay is enhanced with increased $\Omega$a during periods with no entrainment of low-pH waters. However, coral growth must be measured during both seasons in order to confirm this assumption. Threshold values of $\Omega$a when coral growth likely approaches zero were derived from the..."

[RC1] Technical corrections REPLY: All technical corrections will be included

---

## Author Comment (AC2) · 29 Jan 2018

We appreciate all comments and suggestions from anonymous referees. In the following paragraphs we listed our response to the comments from Anonymous Referee #2:

Overall

[RC2] The authors present interesting time-series data of pH and pCO2 from an upwelling region of Pacific Costa Rica. These data are much needed and this contributes to our limited understanding of CO2 dynamics in the eastern Pacific. I do believe there

is a publication nested within this draft, but there will need to be a significant cutting down and revision based on my comments below. The paper will be considerably shortened and changed slightly in scope, but still is publishable.

Big comments

[RC2] 1) The calculation of TA and DIC from pH and pCO2 is unreliable and prone to large errors (see Cullison-Gray et al. 2011; Gray et al. 2011, which you cite). This is probably why you are reporting such unbelievably high values of TA (>2600) and DIC (>2300). Not only do these values never occur on coral reefs (to my knowledge), even in strong upwelling zones like the ETP, you are also reporting relatively low salinities of 32.5. This makes these high values all the more suspect because if you calculate salinity normalized for a TA = 2715, you get NTA= 2923! These types of DIC and TA values are unheard of from reef environments. As such, you need to remove all DIC and TA values that were calculated from pH and pCO2, including the model you generated from some of those values to interpret rates of metabolism driving diurnal changes. Unfortunately, this all has to be deleted from the paper.

REPLY: We understand the comment as follows: Calculations of TA and DIC are prone to errors. Our calculation of unrealistic high TA and DIC values proves this and consequently all calculated data including the model have to be erased from the ms.

First of all we agree to the first statement of Referee #2, that this hardly affects the scope of the paper, which would still be publishable. We appreciate this comment very much.

In the ms we will explain that calculation of TA and DIC from pH and pCO2 are prone to errors and that we will therefore focus on the pH and pCO2 values.

However, the model was produced to explain the observed changes in pH and pCO2 and to untangle the effects of interacting processes on the pCO2 and pH. For example, as mentioned by the reviewer (see comments below) that photosynthesis explains a

drop of pCO2 and increase of pH is correct, but the model also shows that the effect on the pH is insufficient to explain the observed increase of pH in 2013. As indicated by the model, changes in pCO2 and pH can be explained only if carbonate dissolution is considered in addition to photosynthesis. The TA (2120 – 2310) and DIC (1800 – 2040) values used in the model are well within the range reported in other studies (Manzello 2010, Table 1) and far below the extremes, which raised the concern of the reviewer.

In contrast to the pCO2 and the pH, TA and DIC values, which were calculated from the pCO2 and pH, do not reveal a diurnal cycle as the one in the model. This could be a consequence of the errors, which according to results from other studies could range between 16 (Millero 2007, Table 9) and 250 $\mu$mol / kg (Cullison Gray et al. 2011). We will point this out in the ms.

However, the mean DIC and TA values of about 1800 +/- 200 $\mu$mol/kg and 2100 /- 250 $\mu$mol /kg, which were calculated from the pCO2 and pH, are well within the range of those reported from other studies (Manzello 2010, Cyronak et al. 2013) and the one used in the model. This fitting within the range implies, furthermore, that the TA and DIC values used in our model were acceptable.

References: Cullison Gray, S.E., DeGrandpre, M.D., Moore, T.S., Martz, T.R., Friederich, G.E., Johnson, K.S., 2011. Applications of in situ pH measurements for inorganic carbon calculations.Marine Chemistry, 125, 82-90.

Cyronak, T., Santos, I.R., Erler, V.D., Eyre, B.D. 2013. Groundwater and porewater as major sources of alkalinity to a fringing reef lagoon (Muri Lagoon, Cook Islands). Biogeosciences, 10: 2467-2480.

Manzello, D. 2010. Ocean acidification hot spots: spatiotemporal dynamics of the seawater CO2 system of eastern Pacific coral reefs. Limnol. Oceanogr. 55 (1): 239-248.

Millero, F.J. 2007. The marine inorganic carbon cycle. Chem. Rev. 107: 308-341.

[RC2] 2) Please be clear as to what you actually measured versus what you hypothesize, or think is going on. For instance, in the abstract you discuss how calcification is enhanced during the non-upwelling season when saturation states are elevated. This may be true, but at this point it is a hypothesis as you have no data to support it. There are several other instances where hypotheses are discussed as fact, i.e., reef thickness and accretion etc.

REPLY: Writing will be modified through the text, to clarify which variables were measured in the present study and which values were taken from the literature.

[RC2] 3) What is the proximity of the instruments to an actual coral reef? You need to be clear about this because what you are measuring may actually be a result of the metabolism of the carbonate benthos directly under the dock plus the water column processes rather than anything to do with reef dynamics.

REPLY: The instruments were deployed approximately 200 m east of a small Pocillopora spp. patch reef. However, when compared to reefs in the western Pacific Ocean, the reefs in the ETP (e.g. in Bahía Culebra) are poorly developed. Our intention was to characterize the carbonate chemistry within the bay (Bahía Culebra) and its impact on the reefs. Our results indicate that physical oceanographic processes, such as upwelling and exchange between the bay and the open ocean waters, influence the carbonate chemistry on timescales of weeks to months, where the metabolic processes (photosynthesis and calcifications) influence the diurnal cycle. To which extend benthic and pelagic processes control the diurnal cycle, could not be studied based on our data. This will be clarified in the ms.

[RC2] 4) Finally, the paper is generally well written, but would benefit from being proofread by a native English speaker.

REPLY: Thank you. We will take your suggestion into account.

Specific comments

[RC2] P2, line 19: You can't say adaptation without genetics work. Reword to say "adaptation or acclimatization"

REPLY: Will be changed.

[RC2] P2, last line: Thus far there is really no data out there suggesting that OA will directly lead to coral mortality. In fact, in a seminal study, Fine and Tchernov (2007, Science) exposed two coral species to undersaturation and found they stopped growing skeletons and lived in an anemone-like state. Once conditions were raised above saturation, they began producing skeletons again.

REPLY: Accordingly, we will change "coral survival" for "coral reef development within this bay".

[RC2] P3, section 2.2: You discuss taking discrete samples, but do not say how many or how often. Please mention this. The values of TA >2600 and DIC > 2300 have to be calculated from pH-pCO2. You show the result of 8 bottle samples pH relative to the SAMI pH – what about the SAMI CO2 or any other data calculated from the bottles?

REPLY: We could include an additional panel in Fig. 3, showing the regression between SAMI-pCO2 and calculated pCO2 from measured TA and DIC.

[RC2] P5: salinity is a unitless value. PSU describes the scale and is not a unit

REPLY: Accordingly, the term "psu" will be removed from the text.

[RC2] P5, line 5: Need to say what the number after the plus and minus is the first time you use it.

REPLY: The following will be included after the 29.61 $\pm$ 0.93: (average $\pm$ standard deviation)

[RC2] P7, 1st line: why couldn't stimulation of photosynthesis from higher nutrients during upwelling be causing the large amplitude? You can't speak to dissolution without actual TA data and increased photosynthesis seems more likely.

REPLY: See first comment: In principle yes. But even if we would assume that photosynthesis caused the measured decrease of pCO2, the associated impact on the pH would be insufficient to explain the observed increase in pH. The assumption of carbonate dissolution is plausible and would solve this discrepancy. Therefore, we produced the model to test such hypotheses.

[RC2] P7, line 25: Say adapted and or acclimatized.

REPLY: "and/or acclimatized" will be added.

[RC2] P7, last line: Reword to say saturation sate is one of the controlling factors in coral growth. Its likely of secondary or tertiary level importance behind temperature and light.

REPLY: Taking into account your observations and the comments from another reviewer, the full sentence and paragraph will be modified in the following way:

"Aragonite saturation state ($\Omega$a) is known as one of the main variables influencing coral growth and therefore reef distribution around the world (Kleypas et al. 1999). By integrating the data from the present study and values previously reported by Rixen et al. (2012), we estimated that the annual mean $\Omega$a in Bahía Culebra is 3.06. Additionally, earlier studies in the ETP measured $\Omega$a values and coral extension rates from locations that are under the influence of upwelling events (Manzello 2010a), whilst extension rates from Bahía Culebra were measured by Jiménez and Cortés (2003)."

[RC2] P8: delete sentence spanning lines 3 to 5 as well as last sentence of 1st paragraph

REPLY: We will consider removing the sentence spanning lines 3 to 5. However, we consider that the last sentence of the same paragraph is important, as point out that these coral reefs are also threatened by additional stressors resulting from human activities.

[RC2] P8, line 10: Did you measure reef thickness? Also, I'd avoid saying they are

growing on the edge of their ecological tolerance. Its not clear what is meant by that.

REPLY: Reef thickness was visually estimated during numerous dives. We have been working in the study area for more than two decades.

Regarding the second comment, the sentence will be modified: "These reefs are growing in an environment at the limit of reef-building corals tolerance in terms of temperature, nutrient loads and pH (Manzello et al., 2017)"

[RC2] P8, last 5 or so sentences: a lot of conjecture written as fact.

REPLY: Several modifications will be included, in order to reduce conjectures and improve the text:

Original: "Challenging conditions are not restricted to the upwelling season, they occur sporadically also during non-upwelling seasons, when pH and CO2 concentrations reach values comparable to those during upwelling events. Linear extension rates of the main reef building corals in the bay are sensitive to changes in $\Omega$a, suggesting that upwelling reduces coral growth by introducing acidic subsurface waters in the surface layers. Rising levels of $\Omega$a enhance coral growth during the non-upwelling season due to which the linear extension rates of the main reef building corals in Bahía Culebra were among the highest in the ETP; however, reef accretions was low due to erosion. The latter indicates a sensitivity of coral reefs to the intensity of the upwelling and other processes occurring during the non-upwelling/rainy season, such as human impacts on river discharges, the occurrence of cold events (e.g. 2012), and ultimately to OA. Threshold values of $\Omega$a when coral growth likely approaches zero were derived from the correlation of $\Omega$a and linear extension rates and this suggests that OA will seriously threat reefs in Bahía Culebra, which are already at the verge of their ecological tolerance."

Modified: "Challenging conditions for reef development are not restricted to the upwelling season, they occur sporadically also during non-upwelling season, when pH

and CO2 concentrations reach values comparable to those during upwelling events. Previous studies reported that the linear extension rates measured in Bahía Culebra were among the highest in the ETP, thus is likely that coral growth in this bay is enhanced with increased $\Omega a$ during periods with no entrainment of low-pH waters. However, coral growth must be measured during both seasons in order to confirm this supposition. Threshold values of $\Omega a$ when coral growth likely approaches zero were derived from the correlation of $\Omega a$ and previously measured annual linear extension rates. The $\Omega a$ threshold values from the present study and discovery that high-CO2 waters are occasionally hauled into the bay during non-upwelling season; suggest that coral reef development in Bahía Culebra is potentially threatened by anthropogenic OA."

[RC2] Figure 4: your units for pCO2 are incorrect

REPLY: They were also incorrect in figure 5. Both figures will be corrected.

---

## Author Response (AR1)

**1) LIST OF RELEVANT CHANGES MADE IN THE MANUSCRIPT**

1. Abstract was improved following the reviewers suggestions.
2. We modified the abstract, the caption in Fig. 6 and also included additional sentences in sections 2.2 and 4.3, in order to clarify that all coral growth rates and some of the omega values (Panamá and Galápagos) used to evaluate the relationship between carbonate chemistry and coral growth (Fig. 6) were not measured during this study, but taken from previously published data (Jiménez and Cortés, 2003; Manzello, 2010a,b).
3. Methodology used in 2009 for pH and xCO2 measurements was described in more detail (section 2.3).
4. Also in section 2.3, we included all the corresponding values of TA and DIC during periods 2012 and 2013, used as input parameters in the model. We have also included a sentence indicating that despite TA and DIC calculations derived from pH and $p\text{CO}_2$ measurements are prone to errors, our calculated values (including those used as input in the model) are well in range with the values previously reported by other studies.
5. The term rain-ratio was better described, as well as the procedure followed to verify the results from the model. Accordingly, redaction of last sentences in section 2.3 was improved.
6. In section 3.2 we replaced the extreme values of TA and DIC with the mean values, since the values used as input parameters in the model are in range with the mean values for each period.
7. Wording in some sections (particularly in conclusions and section 4.3) was modified, in order to reduce conjectures and improve reading. Sections 4.3 and 5 (conclusions) were highly improved by following all the comments from both reviewers. Modifications included the deletion of some sentences and inclusion of new ones. All these changes can be followed in the marked-up version of the manuscript.
8. An additional panel (b) was included in Fig. 3, in order to show the validation of $p\text{CO}_2$ measurements.
9. Units of $p\text{CO}_2$ in Figs. 4 and 5 were modified. Units of DIC and TA were included in Fig. 5.
10. All technical suggestions from both reviewers were included in the revised version.

**2) POINT-BY-POINT RESPONSE TO REVIEWS**

**Referee #1**

[RC1] First, I think it's important that the authors make it clear at the beginning of the manuscript (Abstract and Methods) that they did not measure coral growth in this study that they used previously published data to evaluate the relationships between carbonate chemistry and coral growth, because this was not obvious to me until the Discussion. I would also say what studies you derived the ecological data from in the Methods and include a brief description of the methodologies those researchers used so that readers can evaluate those data.

REPLY: We have made this point clear in the first sections of the manuscript (Abstract and Methods). The following paragraph was included at the end of section 2.2:

"All coral growth values were taken from the literature; linear extension rates from Bahía Culebra were measured by Jiménez and Cortés (2003), whilst coral growth in Panamá and Galápagos was measured by Manzello (2010a). For the correlation between coral growth and Ωa, we used the mean Ωa values from Panamá and Galápagos previously reported by Manzello (2010b)."

[RC1] The authors should also consider adding some additional text in the Methods/Discussion regarding the collection and interpretation of the carbonate-chemistry data. I've listed a number of specific

comments below, but my most significant concern is with the statement that DIC is temperature-independent. Temperature impacts the solubility of CO2 and directly impacts DIC. Additionally, since the DIC data are derived from pCO2 measurements, which are dependent on T, this does not get rid of the temperature effect on pCO2. I would like to see the author address this issue more thoroughly.

REPLY: The following sentence was included in section 3.2:

"We also compared DIC and TA, in order to estimate to which extend the observedvariations of $p$CO$_2$ were caused by changes in temperature and/or DIC concentrations."

[RC1] P2, L25: But see Toth et al. 2012. Accretion rates in Panama when reefs were growing were comparable to rates observed on reefs in the Caribbean and the authors did not find a significant difference in reef accretion between upwelling and non-upwelling sites. I would suggest making the language here more conservative.

REPLY: The sentence was modified:
Original: "...resulting in poorly cemented coral reefs with low accretion rates that are subject to rapid bioerosion"

Modified: "...and have the potential to produce poorly cemented coral reefs with low accretion rates that are subject to rapid bioerosion"

[RC1] P3, L23: Please provide information on the pH scale used.

REPLY:The pH scale as included => total hydrogen ion scale

[RC1] P3, L28: How can you be sure that the carbonate chemistry at 30 cm was the same as where the instrument was (1.5 m)?

REPLY: We can't. But the samples were taken 30 cm below the surface, which is 1.2 m above the instrument and up to 7.7 m above the bottom (the distance above bottom varied with tide cycles). Since we have never seen any indication of depth related changes (e.g. in turbidity) we assume that DIC and TA changes within the water column are negligible.

[RC1] P4, L8: Did Rixen et al. 2012 use the same methodology for measuring carbonate chemistry? I think it's important to include this information so that readers can fully evaluate the results.

REPLY: The following sentence was included: "In 2009 xCO2 was measured by an underway pCO2 system (SUNDANS) equipped with an infrared gas analyzer (LI-7000), and pH was measured using an Orion ROSS electrode and an Orion Star$^{TM}$."

[RC1] P4, L11-12: I would suggest including the statement "All GLM assumptions were met" here instead of in the Results.

REPLY: It was included as follow : "The GLM was evaluated using graphical methods to identify violations of assumptions of homogeneity of variance and normality of residuals. All GLM assumptions were met."

[RC1] P4, L19: Please include a citation for the rain-ratio (Archer and Maier-Reimer 1994?) and describe what it is.

REPLY: The term rain-ratio seems to be confusing since it actually describes the ratio between the export of organic carbon and calcium carbonate carbon. Accordingly, it was changed as follows:

"$R_{OI}$ describes the ratio between the production of organic carbon (POC) and precipitation of calcium carbonate carbon (PIC), and was used to link $\Delta POC$ to $\Delta PIC$ ($R_{OI}$=POC/PIC) (Eq. 2, 3)."

[RC1] P4, L23: What are the ROI values of -2.6 and 0.8 based on. Please explain in the text where these numbers came from.

REPLY: These numbers were used to run the model in order to obtain the best fit between measured and calculated pH and pCO2 values. The assumption of calcium carbonate dissolution caused the negative sign.

Modified sentence read as follow: "To verify the results from the model, we used the output $\Delta DIC$ and $\Delta TA$ to calculate new pCO2 and pH values, which were further compared to the measured ones. The best fit between modeled and measured values was achieved with a respective $R_{OI}$ of -2.6 for 2012 and 1.0 for 2013, whereas the assumption of calcium carbonate dissolution caused the negative sign."

[RC1] P5, L16-17: Temperature variability also can impact DIC. Also since these data are derived from pCO2 measurements, which are dependent on T, this does not get rid of the temperature effect

REPLY: What we meant to say was that we tried to estimate to which extends temperature and/or changes in the DIC concentrations had an effect on the observedvariations of pCO2. The following sentence was included in section 3.2:

"We also compared DIC and TA, in order to estimate to which extend the observed variations of pCO2 were caused by changes in temperature and /or DIC concentrations."

[RC1] P5, L22: include error term for the aragonite saturation state

REPLY: Standard deviation (SD) was included as follows: 3.06 ± 0.49

[RC1] P6, L5-7: It would be more informative to talk about how much DIC dropped during the "upwelling-like" event in 2012 than to talk about the overall average for the whole sampling period.

REPLY: The following sentence was included: During the seven days that lasted the cold-water intrusion event (June 10-17) the DIC concentrations dropped from 2355.39 µmol kg$^{-1}$ down to 1715.30 µmol kg$^{-1}$.

[RC1] P6, L16: You can't really make conclusions about how regular these sorts of events are based on a few weeks of data from just two years. I would change the wording here to reflect this.

REPLY: Included modification:
Original: "The absence of such a cold-event during the non-upwelling season in 2013 shows that it is an irregular feature…"

Modified: "The absence of such a cold-event during the non-upwelling season in 2013 suggests that the occurrence of this kind of events might be an irregular feature…"

[RC1] P6, L26: The wording of this sentence is confusing. I would rephrase to something like: Although the pCO2 cycles in 2013 followed a similar pattern to 2012, pH was more variable (or less predicable). I would also move this sentence to the end of the previous paragraph and start the next paragraph with "To characterize..."

REPLY:
1) Modified sentence read as follows: "Although the pCO2 cycles in 2013 followed a similar pattern to 2012, pH cycles were less predictable."
2) Sentence was moved to the end of the previous paragraph and the next paragraph started with "To characterize…"

[RC1] P7, L3: I don't understand what this sentence means. Which cycles? Could not be observed in the data? In the model? I actually think that this whole paragraph needs to be fleshed out more. It's not clear to me what the authors are trying to say.

REPLY: The first sentence is certainly confusing, and after a careful reading we concluded that it has no relevance in transmitting the take-home message, which basically is: pH and pCO2 show pronounced daily cycles but the interaction of metabolic processes difficult to identify a clear pattern in the daily cycles of DIC and TA.

To improve reader understands the first sentence of the paragraph was removed. The following paragraph was rephrased and read as follows:

"However, in our case the determined TA and DIC concentrations constrain the impact of the formation of organic matter (POC = photosynthesis – respiration) and calcification (PIC = calcification – dissolution) on the carbonate system. This sets the boundaries within which the observed diurnal cycle of pH and $pCO_2$ has to be explained (Fig. 5c, d). In order to reconstruct the diurnal cycle of pH and $pCO_2$ within these boundaries we assumed a photosynthetic-enhanced calcification during the day and vice versa, dissolution and respiration at night. Thereby the best fit between pH and $pCO_2$ measured in 2013 and the respective calculated values could be obtained by using a $R_{OI}$ of 1. This approach failed to explain the diurnal cycle of pH and $pCO_2$ as observed during the 2012 cold-water intrusion event (June 10-17). The only solution we found to explain these pronounced diurnal cycles within the given DIC and TA boundaries was to assume that photosynthesis and dissolution prevailed during the day and respiration and calcification occurred at night. The $R_{OI}$ of -2.6 resulted in the best fit between the measured and calculated pH and $pCO_2$ for the 2012 event, whereas the negative sign reflects the contrasting effects of calcification and dissolution on the DIC concentration."

[RC1] P7, L10: add a reference to Fig. 5c&d at the end of the sentence ending with "respiration"

REPLY: reference to Fig. 5c and 5d was included.

[RC1] P7, L10: Clarify whether you are describing changes that just happened during the cold-water event or during the entire sampling period in 2012.

REPLY: We are describing the changes during the cold-water event (7 days), not during the full period 2012 (15 days). For a better understanding, the sentence was modified as previously described in [RC1] P7, L3.

[RC1] P7, L30-31: This sentence is misleading. I think that what you mean to say is that saturationstate in each of those locations predicts extension rate. I'm assuming that you used the saturation states/linear extension determined by Manzello 2010 for Panama and Galapagos, but this is not clear in either the text or the figure.

REPLY: The work by Manzello was cited in the text and in Fig. 6. We have also modified the sentenceto clarify that we used those values as input toestimate the correlation between omega and coral growth (Fig. 6).

Writing in text was improved in the following way:

"Aragonite saturation state ($\Omega_a$) is known as one of the main variables influencing coral growth and therefore reef distribution around the world (Kleypas et al. 1999). By integrating the data from the present study and values previously reported by Rixen et al. (2012), we estimated that the annual mean $\Omega_a$ in Bahía Culebra is 3.06. Additionally, earlier studies in the ETP measured $\Omega_a$ values and coral extension rates from locations that are under the influence of upwelling events (Manzello 2010a), whilst extension rates from Bahía Culebra were measured by Jiménez and Cortés (2003). The correlation between our estimated $\Omega_a$ with the available data from Bahía Culebra, Panamá and Galápagos indicates that coral extension rates in each of those locations are predictable by their corresponding $\Omega_a$ values (Fig. 6)."

Figure caption was modified as follow: "Mean aragonite saturation states ($\Omega_a$) - from present and former studies - versus previously reported mean linear extension rates of *Pocilloporadamicornis*and *Pavonaclavus*from upwelling areas in Costa Rica (CR) (Jiménez and Cortés, 2003), Panamá (PAN) and Galápagos (GAL) (Manzello, 2010a). Red broken line shows the regression equation as estimated by Rixen et al. (2012). Red mark represents our estimated $\Omega_a$ threshold for Bahía Culebra, when coral growth equals zero"

[RC1] P8, L11-12: I think it would be good to compare your observation of reef thickness with reef thickness elsewhere in Costa Rica or elsewhere in the ETP, as most reader won't be familiar with how thick a reef "should" be see:

REPLY: P8 L10-12 was modified as follows: "Despite the corals' high annual mean linear extension rates, studies carried out in 1973 showed that the thickness of the reef framework within our study area was with 0.6 to 3 m (mean 1.8 m) among the lowest in the ETP where Holocene framework accumulation in Pocillopora-dominated reefs could reach up to 9 m (Glynn et al., 1983; Toth et al., 2017). During the last decade it further decreased (Alvarado et al., 2012), and during the period of our observation the reef frameworks of Pocillopora spp in Bahía Culebra hardly exceeded a thickness of 0.5 m."

[RC1] P8, L14: It's not correct to talk about "coral" accretion because accretion is a net measure of the growth of a reef. I would re-phrase the beginning of this sentence to something like: Gaps in coral-reef accretion. I would also specify that these gaps are known from the geological record of the ETP. They are not common elsewhere.

REPLY: The sentence was modified:
Original: "Coral's accretional gaps, which are known from the geological records, document the vulnerability of these reefs. They have been..."

Modified: "Gaps in coral reef accretion at the ETP are known from the geological record (Toth et al. 2012; 2015; 2017). They have been..."

[RC1] P8, L20: It's likely that OA will threaten places like Bahia Culebra sooner than elsewhere right? It might be interesting to include a brief discussion of when/how OA may impact reefs in the ETP vs elsewhere and what the implications of this would be.

REPLY: We have included the following sentences just before the last sentence of the current paragraph:

According to climate predictions, the global $\Omega_a$ will reach values < 2.0 by the end of this century (IPCC, 2014) and major upwelling systems such as those off California and South America will intensify (Wang et al., 2015). Combined effects of ocean acidification and impacts of stronger upwelling on $\Omega_a$ in the ETP and on $\Omega_a$ in Bahía Culebra are difficult to predict.

"Worldwide, OA is expected to reduce coral reefs' resilience by decreasing calcification and increasing dissolution and bioerosion (Kleypas et al., 1999; Yates and Halley, 2006a; Anthony et al., 2011). Coral reefs from the ETP are affected by chronic and acute disturbances, such as thermal stress and natural acidification resulting from ENSO and upwelling events, respectively (Manzello et al., 2008; Manzello, 2010b). Historically, these reefs have shown a high resilience to both stressors by separately but their coupled interaction can cause coral reef lost within the next decades. The ETP have the lowest $\Omega_a$ of the tropics, near to the threshold values for coral reef distribution, therefore the reefs from this region may be the most affected by the increasing levels of anthropogenic CO2 and also show the first negative impacts of this human induced OA (Manzello et al., 2017). This emphasizes the importance of the Paris agreement and all the global efforts to reduce the CO2 emission into the atmosphere (Figueres et al., 2017)"

[RC1] P8, L31: I would make it clear here that previous studies have shown that reef accretion is low in Bahia Culebra.

REPLY: We have modified the sentence in section 4.3 (P8, L10-11) to clarify this point; therefore we consider that this statement can be removed from the "Conclusions" section. Sentences spanning lines P8, L28 to P9, L1 were modified as follow:

Original sentence: "Rising levels of $\Omega_a$ enhance coral growth during the non-upwelling season due to which the linear extension rates of the main reef building corals in Bahía Culebra were among the highest in the ETP; however, reef accretion was low due to erosion".

Modified sentence: "Previous studies reported that the linear extension rates measured in Bahía Culebra were among the highest in the ETP, thus is likely that coral growth in this bay is enhanced with increased $\Omega_a$ during periods with no entrainment of low-pH waters. However, coral growth must be measured during both seasons in order to confirm this assumption. Threshold values of $\Omega_a$ when coral growth likely approaches zero were derived from the..."

[RC1] Technical corrections: All technical corrections were included

**Referee #2**
[RC2]The calculation of TA and DIC from pH and pCO2 is unreliable and prone to large errors (see Cullison-Gray et al. 2011; Gray et al. 2011, which you cite). This is probably why you are reporting such unbelievably high values of TA (>2600) and DIC (>2300). Not only do these values never occur on coral reefs (to my knowledge), even in strong upwelling zones like the ETP, you are also reporting relatively low salinities of 32.5. This makes these high values all the more suspect because if you calculate salinity normalized for a TA = 2715, you get NTA= 2923! These types of DIC and TA values are unheard of from

reef environments.As such, you need to remove all DIC and TA values that were calculated from pH and pCO2, including the model you generated from some of those values to interpret rates of metabolism driving diurnal changes. Unfortunately, this all has to be deleted from the paper.

REPLY:
We have included a sentence in section 2.3 explaining that calculation of TA and DIC from pH and pCO2 are prone to errors, which according to results from other studies could range between 16 (Millero 2007, Table 9) and 250 µmol / kg (Cullison Gray et al. 2011). For the same, reason in section 4.1 we focused on the pH and pCO2 values.

However, the model was produced to explain the observed changes in pH and pCO2 and to untangle the effects of interacting processes on the pCO2 and pH. For example, as mentioned by the reviewer (see comments below) that photosynthesis explains a drop of pCO2 and increase of pH is correct, but the model also shows that the effect on the pH is insufficient to explain the observed increase of pH in 2013. As indicated by the model, changes in pCO2 and pH can be explained only if carbonate dissolution is considered in addition to photosynthesis. The TA (2120 – 2310) and DIC (1800 – 2040) values used in the model are well within the range reported in other studies (Manzello 2010, Table 1) and far below the extremes, which raised the concern of the reviewer. To improve the understanding, we have also included in section 2.3 the TA and DIC values used as input parameters in the model and the extreme values were removed from Table 1.

In contrast to the pCO2 and the pH, TA and DIC values, which were calculated from the pCO2 and pH, do not reveal a diurnal cycle as the one in the model. However, the mean DIC and TA values of about 1800 +/- 200 µmol/kg and 2100 /- 250 µmol /kg, which were calculated from the pCO2 and pH, are well within the range of those reported from other studies (Manzello 2010, Cyronak et al. 2013) and the one used in the model. This fitting within the range implies, furthermore, that the TA and DIC values used in our model were acceptable.

[RC2] Please be clear as to what you actually measured versus what you hypothesize, or think is going on. For instance, in the abstract you discuss how calcification is enhanced during the non-upwelling season when saturation states are elevated. This may be true, but at this point it is a hypothesis as you have no data to support it. There are several other instances where hypotheses are discussed as fact.

REPLY: Writing was modified through the text, to clarify which variables were measured in the present study and which values were taken from the literature. Sentences where hypotheses were discussed as facts were removed or modified.

[RC2] What is the proximity of the instruments to an actual coral reef? You need to be clear about this because what you are measuring may actually be a result of the metabolism of the carbonate benthos directly under the dock plus the water column processes rather than anything to do with reef dynamics.

REPLY: The instruments were deployed approximately 200 m east of a small *Pocillopora* spp. patch reef. However, when compared to reefs in the western Pacific Ocean, the reefs in the ETP (e.g. in Bahía Culebra) are poorly developed. Our intention was to characterize the carbonate chemistry within the bay (Bahía Culebra) and its impact on the reefs. Our results indicate that physical oceanographic processes, such as upwelling and exchange between the bay and the open ocean waters, influence the carbonate chemistry on timescales of weeks to months, where the metabolic processes (photosynthesis and calcifications) influence the diurnal cycle. To which extend benthic and pelagic processes control the diurnal cycle, could not be studied based on our data. This was clarified in the conclusions section.

[RC2] P2, line 19: You can't say adaptation without genetics work. Reword to say "adaptation or acclimatization"

REPLY: Was changed.

[RC2] P2, last line: Thus far there is really no data out there suggesting that OA will directly lead to coral mortality. In fact, in a seminal study, Fine and Tchernov (2007, Science) exposed two coral species to undersaturation and found they stopped growing skeletons and lived in an anemone-like state. Once conditions were raised above saturation, they began producing skeletons again.

REPLY: Accordingly, we changed "coral survival" for "coral reef development within this bay".

[RC2] P3, section 2.2: You discuss taking discrete samples, but do not say how many or how often. Please mention this. The values of TA >2600 and DIC > 2300 have to be calculated from pH-pCO2. You show the result of 8 bottle samples pH relative to the SAMI pH – what about the SAMI CO2 or any other data calculated from the bottles?

REPLY: We included an additional panel in Fig. 3, showing the regression between SAMI-pCO2 and calculated pCO2 from measured TA and DIC. Due to logistics the frequency for collection of discrete water samples was irregular, but in section 2.2 was pointed out that they were collected as often as possible.

[RC2] P5: salinity is a unitless value. PSU describes the scale and is not a unit

REPLY: Accordingly, the term "psu" was removed from all the text.

[RC2] P5, line 5: Need to say what the number after the plus and minus is the first time you use it.

REPLY: The following was included after the 29.61 ± 0.93: (average ± standard deviation)

[RC2] P7, 1st line: why couldn't stimulation of photosynthesis from higher nutrients during upwelling be causing the large amplitude? You can't speak to dissolution without actual TA data and increased photosynthesis seems more likely.

REPLY: See first comment: In principle yes. But even if we would assume that photosynthesis caused the measured decrease of pCO2, the associated impact on the pH would be insufficient to explain the observed increase in pH. The assumption of carbonate dissolution is plausible and would solve this discrepancy. Therefore, we produced the model to test such hypotheses.

[RC2] P7, line 25: Say adapted and or acclimatized.

REPLY: "and/or acclimatized" was added.

[RC2] P7, last line: Reword to say saturation sate is one of the controlling factors in coral growth. Its likely of secondary or tertiary level importance behind temperature and light.
REPLY: Taking into account your observations and the comments from another reviewer, the full sentence and paragraph was modified (see reply to RC1 P7, L30-31).

[RC2] P8: delete sentence spanning lines 3 to 5 as well as last sentence of 1st paragraph

REPLY: Sentence spanning lines 3 to 5 was removed. However, we consider that the last sentence of the same paragraph is important, as point out that these coral reefs are also threatened by additional stressors resulting from human activities, therefore was not deleted from the text.

[RC2] P8, line 10: Did you measure reef thickness? Also, I'd avoid saying they are growing on the edge of their ecological tolerance. Its not clear what is meant by that.

REPLY: Reef thickness was visually estimated during numerous dives. We have been working in the study area for more than two decades.

Regarding the second comment, the sentence was modified: "These reefs are growing in an environment at the limit of reef-building corals tolerance in terms of temperature, nutrient loads and pH (Manzello et al., 2017)"

[RC2] P8, last 5 or so sentences: a lot of conjecture written as fact.

REPLY: Several modifications were included, in order to reduce conjectures and improve the text:

Original: "Challenging conditions are not restricted to the upwelling season, they occur sporadically also during non-upwelling seasons, when pH and CO2 concentrations reach values comparable to those during upwelling events. Linear extension rates of the main reef building corals in the bay are sensitive to changes in Ωa, suggesting that upwelling reduces coral growth by introducing acidic subsurface waters in the surface layers. Rising levels of Ωa enhance coral growth during the non-upwelling season due to which the linear extension rates of the main reef building corals in Bahía Culebra were among the highest in the ETP; however, reef accretions was low due to erosion. The latter indicates a sensitivity of coral reefs to the intensity of the upwelling and other processes occurring during the non-upwelling/rainy season, such as human impacts on river discharges, the occurrence of cold events (e.g. 2012), and ultimately to OA. Threshold values of Ωa when coral growth likely approaches zero were derived from the correlation of Ωa and linear extension rates and this suggests that OA will seriously threat reefs in Bahía Culebra, which are already at the verge of their ecological tolerance."

Modified: "Challenging conditions for reef development are not restricted to the upwelling season, they occur sporadically also during non-upwelling season, when pH and CO2 concentrations reach values comparable to those during upwelling events. Previous studies reported that the linear extension rates measured in Bahía Culebra were among the highest in the ETP, thus is likely that coral growth in this bay is enhanced with increased Ωa during periods with no entrainment of low-pH waters. However, coral growth must be measured during both seasons in order to confirm this supposition. Threshold values of Ωa when coral growth likely approaches zero were derived from the correlation of Ωa and previously measured annual linear extension rates. The Ωa threshold values from the present study and the fact that high-CO2 waters are occasionally hauled into the bay during non-upwelling season; suggest that coral reef development in Bahía Culebra is potentially threatened by anthropogenic OA."

[RC2] Figure 4: your units for pCO2 are incorrect

REPLY: They were also incorrect in figure 5. Both figures were corrected.

**3) MARKED-UP VERSION**

**Natural ocean acidification at Papagayo upwelling system (North Pacific Costa Rica): implications for reef development**

Celeste Sánchez-Noguera[1,2], Ines Stuhldreier[1,3], Jorge Cortés[2], Carlos Jiménez[4,5], Álvaro Morales[2,6], Christian Wild[3], Tim Rixen[1,7]

[1]Leibniz Centre for Tropical Marine Research (ZMT), Bremen, D-28359, Germany
[2]Centro de Investigación en Ciencias del Mar y Limnología (CIMAR), San José, 11501-2060, Costa Rica
[3]Faculty of Biology and Chemistry (FB2), University of Bremen, Bremen, 28359, Germany
[4]Energy, Environment and Water Research Center (EEWRC) of the Cyprus Institute (CyI), Nicosia, 1645, Cyprus
[5]Enalia Physis Environmental Research Centre (ENALIA), Aglanjia, 2101, Nicosia, Cyprus
[6]Escuela de Biología, University of Costa Rica, San José, Costa Rica
[7]Institute of Geology, University Hamburg, Hamburg, 20146, Germany

*Correspondence to*: Celeste Sánchez-Noguera (celeste08@gmail.com)

**Abstract.** Numerous experiments have shown that ocean acidification impedes coral calcification, but knowledge about in situ reef ecosystem response to ocean acidification is still scarce. Bahía Culebra, situated at the northern Pacific coast of Costa Rica, is a location naturally exposed to acidic conditions due to the Papagayo seasonal upwelling. We measured pH and $pCO_2$ in situ during two non-upwelling seasons (June 2012, May-June 2013), with a high temporal resolution of every 15 and 30 min, respectively, using two Submersible Autonomous Moored Instruments (SAMI-pH, SAMI-CO2). These results were compared with published data from the 2009 upwelling season.  Findings revealed that the carbonate system in Bahía Culebra shows a high temporal variability. Incoming offshore waters drive inter- and intra-seasonal changes. Lowest pH (7.8) and highest $pCO_2$ (658.3 µatm) values measured during a cold-water intrusion event in the non-upwelling season were similar to those minimum values reported from upwelling season (pH = 7.8, $pCO_2$ = 643.5 µatm), unveiling that natural acidification also occurs sporadically  in the non-upwelling season. This affects the interaction of photosynthesis, respiration, calcification, and carbonate dissolution and the resulting diel cycle of pH and $pCO_2$ in the reefs of Bahía Culebra. During non-upwelling season the aragonite saturation state ($\Omega_a$) rises to values of >3.3 and  during upwelling season fall below 2.5. The $\Omega_a$ threshold values for coral growth were derived from the correlation between measured $\Omega_a$ and coral linear extension rates which were obtained from the literature and suggest that future ocean acidification will threaten the continued growth of 
[revised manuscript text omitted]
. All coral growth values were taken from the literature; linear extension rates from Bahía Culebra were measured by Jiménez and Cortés (2003), whilst coral growth in Panamá and Galápagos was measured by Manzello (2010a). For the correlation between coral growth and $\Omega_a$, we used the mean $\Omega_a$ values from Panamá and Galápagos previously reported by Manzello (2010b).

10   **2.3 Data analysis**

We compared our data with values measured during upwelling season in 2009 (Rixen et al., 2012). In 2009 $x$CO$_2$ was measured by an underway $p$CO$_2$ system (SUNDANS) equipped with an infrared gas analyzer (LI-7000), and pH was measured using an Orion ROSS electrode an Orion Star$^{TM}$. Correlations between tidal cycles and physicochemical parameters (pH, $p$CO$_2$, T, wind) during non-upwelling periods were tested via Pearson Correlation in Python. Differences in
15   parameters (temperature, pH, $p$CO$_2$, TA, DIC, $\Omega_a$) between all periods (2009, 2012, 2013) were tested with a General Linear Model (GLM), in the statistical package R. The GLM was evaluated using graphical methods to identify violations of assumptions of homogeneity of variance and normality of residuals. All GLM assumptions were met. Additionally, we developed a simple model to improve our understanding of processes controlling the observed diel trends, as seen in the time series data of pH and $p$CO$_2$ (Fig. 2, 4). The model simulates combined effects of metabolic processes (photosynthesis,
20   respiration, calcification and dissolution) on the carbonate chemistry. Input parameters for starting the model were the calculated DIC (in 2012: 2037 µmol kg$^{-1}$ at 7:00 h and 2019 µmol kg$^{-1}$ at 15:00 h; in 2013: 1883 µmol kg$^{-1}$ at 5:00 h and 1805 µmol kg$^{-1}$ at 15:00 h) and TA (in 2012: 2284 µmol kg$^{-1}$ at 7:00 h; in 2013: 2193 µmol kg$^{-1}$ at 5:00) values, corresponding to the highest and lowest measured $p$CO$_2$ during the day. Calculation of TA and DIC from the pair pH and $p$CO$_2$ is prone to errors (Millero, 2007; Cullison Gray et al., 2011), however the values used as input parameters in the model
25   are in range with those reported from other studies in tropical areas (Manzello, 2010b; Cyronak et al., 2013b). DIC and TA values corresponding to the highest and lowest measured $p$CO$_2$ during day and night. The difference between the two DIC concentrations (ΔDIC) was assumed to be caused by photosynthesis/respiration and the resulting formation and decomposition of particulate organic carbon (POC) as well as calcification/dissolution and the precipitation and dissolution of particulate inorganic carbon (PIC, Eq. 1). The rain ratio ($R_{OI}$ describes the ratio between the production of organic carbon
30   (POC) and precipitation of calcium carbonate carbon (PIC), and =POC/PIC) was used to link ΔPOC to ΔPIC ($R_{OI}$=POC/PIC) (Eq. 2, 3). The $R_{OI}$ was further constrained by the determined change of TA (ΔTA).(ΔTA). Therefore, it was considered that photosynthesis and /respiration of one mole of carbon increases and reduces TA by 0.15 units, respectively

(Broecker and Peng, 1982). Calcification and /dissolution of one mole of carbon decreases and increases TA by two units (Eq. 4). To $_{OI}$ verify the  results from the model, we used the output  ΔDIC and ΔTA  to calculate new $p\text{CO}_2$ and pH values, which were further compared to the measured ones (Fig. 5). The best fit between modeled and measured values was achieved with a respective $R_{OI}$ of -2.6 for 2012 and 1.0 for 2013, whereas the assumption of calcium carbonate dissolution caused the negative sign.

$$\Delta DIC = \Delta POC + \Delta PIC \tag{1}$$

$$\Delta PIC = \left(\frac{\Delta POC}{R_{OI}}\right) \tag{2}$$

$$\Delta POC = \Delta DIC / \left(1 + \left(\frac{1}{R_{OI}}\right)\right) \tag{3}$$

$$\Delta TA = (\Delta POC * 0.15) - \left(\left(\frac{\Delta POC}{R_{OI}}\right) * 2\right) \tag{4}$$

This was calculated on hourly time steps, separately for 2012 and 2013, using the mean SWT (2012 = 29.61 ± 0.93 °C, 2013 = 30.08 ± 0.27 °C) and salinity (2012 = 32.5 2013 = 32.5)

**3 Results**

**3.1 Carbonate chemistry during non-upwelling season**

In June 2012, average SWT was 29.61 ± 0.93 (average ± standard deviation) °C and ranged from 27.13 °C to 31.37 °C. In May-June 2013 SWT ranged from 29.3 °C to 30.7 °C (average 30.08 ± 0.27°C). During both periods, the salinity was 32.5 ± 0.8. During the study periods, the wind intensified during the afternoons reaching speeds of up to 8.5 m s$^{-1}$ and 6.0 m s$^{-1}$ in 2012 and 2013, respectively (Fig. 2). Average pH and $p\text{CO}_2$ in June 2012 were 7.98 ± 0.04 and 456.38 ± 69.68 µatm, respectively; the corresponding averages for May-June 2013 were 8.02 ± 0.03 and 375.67 ± 24.25 µatm. Since the tidal cycle was not significantly correlated with the variability of pH, $p\text{CO}_2$, T or wind (p > 0.05) during the periods of observations (Table 2), it was excluded from further discussions. Mean $\Omega_a$ values were 3.32 ± 0.46 in June 2012 and 3.50 ± 0.49 in May-June 2013 (Table 1).

**3.2 Seasonal variation of the carbonate system**

Measured parameters showed significant differences between study periods (p < 0.05). The SWT range differed among years (Table 1); 2013 was the warmest study period, followed by 2012 and 2009. Lowest measured pH was 7.81 in June 2012, 7.84 in April 2009 and 7.95 in May-June 2013. We also compared DIC and TA, in order to estimate to which extend the observed variations  of $p\text{CO}_2$

were caused by changes in temperature and/or DIC concentrations. Mean. Highest DIC values were 2098.71 ± 103.812360.92 µmol kg$^{-1}$ in April 2009, 1924.65 ± 195.072355.39 µmol kg$^{-1}$ in June 2012 and 1800.92 ± 142.782199.50 µmol kg$^{-1}$ in May-June 2013. Similarly, meanhighest TA values were 2328.42 ± 118.452714.84 µmol kg$^{-1}$ in June 2102, 2610.47 µmol kg$^{-1}$ in April 2009, 2204.54 ± 212.18 µmol kg$^{-1}$ in June 2012 and 2102.66 ± 174.79 and 2599.99 µmol kg$^{-1}$ in May-June 2013. According to average values, April 2009 was the period with most acidic water and greater $CO_2$ enrichment, followed by June 2012 and May-June 2013 (Table 1). Mean $\Omega_a$ values were 2.71 ± 0.29 during upwelling season (April 2009) and 3.413.37 ± 0.130.47 during non-upwelling season (June 2012, May-June 2013), resulting in an annual average $\Omega_a$ of 3.06 ± 0.493.04 at Bahía Culebra. Time series of pH and $p$CO$_2$ in June 2012 and May-June 2013 showed a pronounced daily cycle (Fig. 4), which in addition to previously described data will be discussed in the following paragraphs.

**4 Discussion**

**4.1 Natural OA beyond the upwelling season**

The observed differences in pH and $p$CO$_2$ between 2012 and 2013 suggest that the non-upwelling season exhibits a strong interannual variability (Table 1). In 2012 pH was lower and $p$CO$_2$ higher than in 2013 (Fig. 2b, c). The June 2012 time-series data showed that SWT decreased and $p$CO$_2$ increased from 300 to 650 µatm in less than a week, after several days of strong afternoon winds (Fig. 2a). Similarly, this increase in $p$CO$_2$ was accompanied by a dropped in pH form 8.04 to 7.83 (Fig. 2a). This suggestsThis suggests, that an enhanced wind-driven vertical mixing entrained cooler and $CO_2$-enriched waters from greater water-depth into the surface layer. The associated SWT drop from 31.4 °C to 27.1 °C was similar to thatthose observed during the onset of the 2009 upwelling event (26.2 °C to 23.7 °C; Rixen et al., 2012). Nevertheless, the higher SWT during the 2012 non-upwelling season suggests that the entrained water originated from a shallower water-depth, compared withto the water upwelled in 2009. The $p$CO$_2$ values with up to 650 µatm reached the same level during both events, which is partially caused by the higher SWT in 2012. However, the temperature independent DIC concentrations in 2012 (1924.65 ± 195.07 µmol kg$^{-1}$) were lower thanfell below those in 2009 (2098.71 ± 103.81 µmol kg$^{-1}$), but exceeded those in 2013 (1800.92 ± 142.78 µmol kg$^{-1}$, Table 1). During the seven days that lasted the cold-water intrusion event in 2012 (June 10-17), the DIC concentrations dropped from 2355.39 µmol kg$^{-1}$ down to 1715.30 µmol kg$^{-1}$. This implies that in addition to high SWT, the entrainment of $CO_2$-enriched subsurface water increased the $p$CO$_2$ not only during the upwelling periods, but also during the 2012 non-upwelling season.

Since in 2012 the $p$CO$_2$ had increased already increased by June 7$^{th}$ and the SWT decreased only two days later (June 10$^{th}$), the inflow of $CO_2$-enriched waters seems to have increased the $p$CO$_2$ already prior to the strengthening of local winds (Fig. 2b). Later, local wind-induced vertical mixing seems to have amplified the impact of the inflowing $CO_2$-enriched water mass on the $p$CO$_2$ in the surface water by increasing its input into surface layers. Accordingly, the $CO_2$-enriched waters were apparently supplied from a different location than they are during upwelling season.somewhere else. Since the NECC carries offshore waters towards the Costa Rican shore during the non-upwelling season (Wyrtki, 1965, 1966; Fiedler, 2002), it is

assumed that the CO₂-enriched subsurface water originated somewhere south of our study area in the open ETP. The absence of such a cold-event during the non-upwelling season in 2013 suggests that the occurrence of this kind of events might be an irregular feature (Fig. 2c, d), and the driving forces are still elusive. Nevertheless, these types of events have the potential to affect the metabolic processes in the bay as will be discussed in the following section, which

5    analyzes the daily cycles during the non-upwelling seasons in 2012 and 2013.

**4.2 Processes behind the variability of the carbonate system**

In 2012, the pH and the $p$CO₂ values followed a pronounced diurnal cycle with highest pH and lowest $p$CO₂ values during the late afternoon and lowest pH and highest $p$CO₂ values around sunrise in the early morning (Fig. 4a). Such daily cycles are typical for tropical regions and are assumed to be caused by photosynthesis during the day and respiration of organic

10   matter during the night (Shaw et al., 2012; Albright et al., 2013; Cyronak et al., 2013a). The aragonite saturation state as well as the DIC/TA ratio followed this pattern, with higher $\Omega_a$ and lower DIC/TA ratio values during the day as well as lower $\Omega_a$ and higher DIC/TA values at night (Fig. 4b). Although the $p$CO₂ cycles in 2013 followed a similar pattern to 2012, pH cycles were less predictable (Fig. 4).

15    To characterize the relative importance of the processes responsible for the observed changes in pH and $p$CO₂ (photosynthesis, respiration, calcification and dissolution) we used the model described earlier, which is based on the determined DIC concentrations during times when pH and $p$CO₂ revealed their daily minima and maxima, respectively. For example, if photosynthesis of organic matter dominates the transition from early morning maxima of $p$CO₂ to late afternoon minima of $p$CO₂ it should be associated with a decline in DIC. Whether photosynthesis was accompanied with

20   enhanced calcification can be detected by an associated decrease of TA. Since decreasing DIC raises the pH and a decrease in TA lowers the pH, such photosynthetic enhanced calcification hardly affects the pH and could explain the weak daily cycle observed in 2013. Alternatively, if photosynthesis is accompanied by carbonate dissolution during the day, this would amplify the daily cycle of pH and $p$CO₂ as seen during the cold-water intrusion event in 2012. Likewise, an increased photosynthesis resulting from higher nutrient concentrations (Pennington et al., 2006) could also be causing the observed

25   large amplitude during the event in 2012. However, in our case the determined TA and DIC concentrations constrain the impact of the formation of organic matter (POC = photosynthesis – respiration) and calcification (PIC = calcification – dissolution) on the carbonate system. This sets the boundaries within which the observed diurnal cycle of pH and $p$CO₂ has to be explained (Fig. 5c, d). In order to reconstruct the diurnal cycle of pH and $p$CO₂ within these boundaries we assumed a photosynthetic-enhanced calcification during the day and vice versa, dissolution and respiration at night. Thereby the best fit

30   between pH and $p$CO₂ measured in 2013 and the respective calculated values could be obtained by using a $R_{OI}$ of 1. This approach failed to explain the diurnal cycle of pH and $p$CO₂ as observed during the 2012 cold-water intrusion event (June 10-17). The only solution we found to explain these pronounced diurnal cycles within the given DIC and TA boundaries was to assume that photosynthesis and dissolution prevailed during the day and respiration and calcification occurred at night.

The $R_{OI}$ of -2.6 resulted in the best fit between the measured and calculated pH and $p$CO$_2$ for the 2012 event, whereas the negative sign reflects the contrasting effects of calcification and dissolution on the DIC concentration.

These daily cycles in pH and $p$CO$_2$ suggest concordant cycles in DIC and TA, which unexpectedly could not be observed. The interplay of all the four metabolic processes of relevance (photosynthesis, respiration, calcification and dissolution) seems to be softening the daily cycles of DIC and TA. However, thecycles of DIC and TA, but the pronounced daily cycle of the DIC/TA ratio  indicate that some of these processes control the daily cycles of pH and $p$CO$_2$ (Fig. 4).

~~To identify the dominant processes, we developed a numerical model and recalculated the daily trends of pH and $p$CO$_2$ (Fig. 5). The calculated $p$CO$_2$ and pH agree quite well to the measured ones and support our previous interpretations that during 2013, photosynthesis and light enhanced calcification prevailed during the day and carbonate dissolution was relegated to night hours alongside respiration (Fig. 5c, d). Based on the observed daily pH and $p$CO$_2$ cycles and in line with our model results, it seems that during.(June 10-17), the diurnal cycle of the carbonate, the system was dominated by photosynthesis and calcium carbonate dissolution during light hours, with respiration and calcification occurring at night.particularDark calcification~~ is not entirely uncommon and occurs in both, sandy bottoms and coral reefs (Yates and Halley, 2006b; Albright et al., 2013). Accordingly, the entrainment of CO$_2$-enriched water from the NECC seems to shift the carbonate chemistry of Bahía Culebra from a system where photosynthesis and calcification are the controlling processes during light hours to a system in which daytime is dominated by photosynthesis and dissolution. The net effect, as observed, is an enhanced $p$CO$_2$ and lower $\Omega_a$ during periods characterized by the inflow of CO$_2$-enriched waters (Table 1). This has strong ecological implications for local coral reef ecosystems.

**4.3 Ecological implications for coral reefs**

Coral reefs in Bahía Culebra were dominated by *Pocillopora* spp. and *Pavona clavus* (Jiménez, 2001; Jiménez et al., 2010), whereas *Porites lobata* is the main reef forming coral in the southern part of the Costa Rican Pacific coast (Cortés and Jiménez, 2003; Glynn et al., 2017). Although the reefs in the north are naturally exposed to periodic high-CO$_2$ conditions during upwelling events (Rixen et al., 2012), as well as during cold water intrusions in non-upwelling season, the linear extension rates of *Pocillopora* spp. and *P. clavus* exceeded those of the same species in other regions (Fig. 6) (Glynn, 1977; Jiménez and Cortés, 2003; Manzello, 2010a; Rixen et al., 2012). This suggests that local corals are adapted and/or acclimatized to the upwelling of cold and acidic waters.

Aragonite saturation state ($\Omega_a$) is known as one of the main variables influencing coral growth and therefore reef distribution around the world (Kleypas et al. 1999). By integrating the data from the present study and values previously reported by Rixen et al. (2012), we estimated that the annual mean $\Omega_a$ in Bahía Culebra is 3.06. Additionally, earlier studies in the ETP

measured $\Omega_a$ values and coral extension rates from locations that are under the influence of upwelling events (Manzello 2010a), whilst extension rates from Bahía Culebra were measured by Jiménez and Cortés (2003). The correlation between our estimated $\Omega_a$ with the available data from Bahía Culebra, Panamá and Galápagos indicates that coral extension rates in each of those locations are predicted by their corresponding $\Omega_a$ values (Fig. 6).

5 ~~Since it is generally assumed that $\Omega_a$ strongly influences coral growth, we calculated the annual mean $\Omega_a$ of 3.04 for Bahía Culebra by integrating measurements from all periods (Table 1). This average $\Omega_a$ correlates well with the extension rates of *Pocillopora damicornis* and *P. clavus* from three upwelling influenced areas in the ETP and supports the general assumption that $\Omega_a$ controls the growth of corals (Fig. 6).~~

The dependency of coral growth on $\Omega_a$ and the mean $\Omega_a$ (2.71) during the upwelling season (Table 1) suggests that upwelling

10 of acidic waters should reduce corals' relatively high annual mean growth rates in Bahía Culebra. The increased $\Omega_a$ during non-upwelling season in turn must enhance linear extension and explains corals' high annual mean growth rates.  The $\Omega_a$ values from this study suggest that most favourable conditions for coral growth occur during non-upwelling season, the period

15 that coincides with development of the rainy season. This implies that during the main growing season the eutrophication and siltation caused by human impacts on river discharges, as well as the development of harmful algal blooms, could also strongly affect the corals' annual mean growth rates (Cortés and Reyes-Bonilla, 2017).

Despite the corals' high annual mean linear extension rates, studies carried out in 1973 showed that the thickness of the reef framework within our study area was with 0.6 to 3 m (mean 1.8 m) among the lowest in the ETP, where Holocene

20 framework accumulation in *Pocillopora*-dominated reefs could reach up to 9 m (Glynn et al., 1983; Toth et al., 2017).  During the last decade it further decreased (Alvarado et al., 2012), and during the period of our observation the reef frameworks of *Pocillopora* spp. in Bahía Culebra hardly exceeded a thickness of 0.5 m . This denotes that although *Pocillopora* spp. and *P. clavus* are adapted to the entrainment of acidic waters, these reefs are growing in an environment at the limit of reef-building

25 corals tolerance in terms of temperature, nutrient loads and pH (Manzello et al., 2017). Gaps in coral reef accretion at the ETP are known from the geological record (Toth et al., 2012; 2015; 2017). They have been linked to increased ENSO variability (Toth et al., 2012, 2015) and stronger upwelling conditions (Glynn et al., 1983), favouring dissolution and erosion of reef frameworks while at the same time restricting coral growth.

30 The y-intercept of the regression equation derived from the correlation between linear extension rates and $\Omega_a$ furthermore implies that linear extension of *P. damicornis* and *P. clavus* should approach zero under a carbonate saturation state of $\Omega_a < 2.5$ (*P. damicornis*) and $< 2.2$ (*P. clavus*). According to climate predictions, the global $\Omega_a$ will reach values $< 2.0$ by the end of this century (IPCC, 2014), and major upwelling systems such as those off California and South America will intensify (Wang et al., 2015). Combined effects of ocean acidification and impacts of stronger upwelling on $\Omega_a$ in the

ETP and on $\Omega_a$ in Bahía Culebra are difficult to predict. Worldwide, OA is expected to reduce coral reefs' resilience by decreasing calcification and increasing dissolution and bioerosion (Kleypas et al., 1999; Yates and Halley, 2006a; Anthony et al., 2011). Coral reefs from the ETP are affected by chronic and acute disturbances, such as thermal stress and natural ocean acidification resulting from ENSO and upwelling events, respectively (Manzello et al., 2008; Manzello, 2010b). Historically, these reefs have shown a high resilience to both stressors by separately but their coupled interaction can cause coral reef lost within the next decades. The ETP have the lowest $\Omega_a$ of the tropics, near to the threshold values for coral reef distribution, therefore the reefs from this region may be the most affected by the increasing levels of anthropogenic $CO_2$ and also show the first negative impacts of this human induced OA (Manzello et al., 2017). This emphasizes the importance of the Paris agreement and all the global efforts to reduce the $CO_2$ emission into the atmosphere (Figueres et al., 2017).

**5 Conclusions**

The present study provides data from in situ measurements from a system that is naturally exposed to low-pH conditions, and seeks to characterize the carbonate chemistry within a bay (Bahía Culebra) and its potential impact on the reefs. This study builds on previous field studies in the upwelling areas of Panamá (Manzello et al., 2008; Manzello, 2010b) and Papagayo (Rixen et al., 2012). Our results indicate that physical processes, such as the coastal upwelling and the exchange of water between the bay and the open ocean, influence the carbonate chemistry on timescales of weeks to months, where metabolic processes (photosynthesis and calcification) influence the diurnal cycle. To which extend benthic and pelagic processes control the diurnal cycle, cannot be established based on our data. However, the results from the present study also  suggest that coral reefs from Bahía Culebra are exposed to a high intra- and interannual variability in the carbonate system. Challenging conditions for reef development are not restricted to the upwelling season, they occur sporadically also during non-upwelling season, when pH and $CO_2$ concentrations reach values comparable to those during upwelling events. Previous studies reported  that  the linear extension rates measured in Bahía Culebra were among the highest in the ETP, thus is likely that coral growth in this bay is enhanced with increased $\Omega_a$ during periods with no entrainment of  low-pH waters. However, coral growth must be measured  during both seasons in order to confirm this assumption. Threshold values of $\Omega_a$ when coral growth likely approaches zero were derived from the correlation of $\Omega_a$ and previously measured annual linear extension rates. The $\Omega_a$ threshold values from the present study and the fact that high-$CO_2$ waters are occasionally hauled in to the bay during non-upwelling season; suggest that coral

reef development in Bahía Culebra is potentially threatened by anthropogenic OA.

**6 Data availability**

Data are available by direct request to the corresponding author.

**7 Author contribution**

C. Sánchez-Noguera and T. Rixen designed the study, analyzed the data, prepared figures and/or tables and wrote the paper C. Sánchez-Noguera collected and analyzed the samples. I. Stuhldreier, J. Cortés, Á. Morales, C. Jiménez and C. Wild reviewed the paper.

**8 Competing interests**

The authors declare that they have no conflict of interest.

**9 Disclaimer**

This study was funded by the Leibniz Association, as part of the PhD research of C. Sánchez-Noguera. Funders had no role in conceiving the study, collection and analysis of data or manuscript preparation.

**10 Acknowledgements**

This project was conducted in cooperation with the Centro de Investigación en Ciencias del Mar y Limnología (CIMAR), University of Costa Rica. Special thanks to Marina Papagayo for allowing us to deploy the sensors in their facilities, Giovanni Bassey and Carlos Marenco for logistic support and sample collection.

**Table 1: Measured and calculated (\*) parameters, during upwelling (2009) and non-upwelling seasons (2012, 2013) at Bahía Culebra, Costa Rica.**

| | pH (Total scale) | $p\mathrm{CO_2}$ ($\mu$atm) | $CO_2$ ($\mu$mol kg$^{-1}$) | T (°C) | DIC* ($\mu$mol kg$^{-1}$) | TA* ($\mu$mol kg$^{-1}$) | $\Omega$* |
|---|---|---|---|---|---|---|---|
| **2009** | | | | | | | |
| Mean ± SD | 7.91 ± 0.32 | 578.49 ± 42.82 | 16.44 ± 1.35 | 25.09 ± 0.57 | 2098.71 ± 103.81 | 2328.42 ± 118.45 | 2.71 ± 0.29 |
| **2012** | | | | | | | |
| Mean ± SD | 7.98 ± 0.04 | 456.38 ± 69.68 | 11.77 ± 1.99 | 29.61 ± 0.93 | 1924.65 ± 195.07 | 2204.54 ± 212.18 | 3.32 ± 0.46 |
| **2013** | | | | | | | |
| Mean ± SD | 8.02 ± 0.03 | 375.67 ± 24.25 | 9.56 ± 0.64 | 30.08 ± 0.27 | 1800.92 ± 142.78 | 2102.66 ± 174.79 | 3.50 ± 0.49 |

**Table 2: Correlations between tide height and four parameters during non-upwelling season (2012, 2013).**

| Year | pH | $p\text{CO}_2$ | T | Wind |
|------|--------|-------|--------|--------|
| 2012 | -0.004 | 0.037 | -0.005 | 0.033 |
| 2013 | 0.111 | 0.026 | -0.093 | -0.126 |

All p-values > 0.05

[Figure]

**Figure 1: Location of Bahía Culebra (square) in the Gulf of Papagayo, North Pacific coast of Costa Rica (insert). Measurements were made at Marina Papagayo (star). Main ocean currents influencing the Gulf of Papagayo (dashed arrows): NECC= North Equatorial Counter Current, CRCC= Costa Rica Coastal Current.**

[Figure]

**Figure 2: Measured parameters (wind speed, SWT, pH and $p$CO$_2$) during the non-upwelling seasons of June 2012 (a, b) and May-June 2013 (c, d), at Bahía Culebra. Shaded area in (a) and (b) indicates the 2012 upwelling-like event.**

[Figure]

**Figure 3: Validation of   using discrete  samples.  measured with a VINDTA 3C system.**

[Figure]

**Figure 4: Diel pattern of parameters measured in Bahía Culebra. Data points are hourly averages of 15 and 7 consecutive days in 2012 (a, b) and 2013 (c, d), respectively. The shaded area represents daylight hours.**

[Figure]

**Figure 5:** Expected diel behavior of the carbonate system in 2012 (a, b) and 2013 (c, d), based on measured parameters. Modeled parameters are shown as blue crosses and empty circles, the reference parameter used to adjust the model is shown in black triangles. Shaded area represents daylight hours.

[Figure]

**Figure 6: Mean aragonite saturation state ($\Omega_a$)  versus  mean linear extension rates of  *Pocillopora damicornis* and  *Pavona clavus*  from upwelling areas in Costa Rica (CR)  Panamá (PAN) and Galápagos (GAL)  Manzello, 2010a). Red broken line shows the regression equation  estimated by Rixen et al. (2012), the red mark represents our estimated $\Omega_a$  for Bahía Culebra when coral growth equals zero.**